# Development and parameter estimation of snow-melt models using spatial snow-cover observations from MODIS

Dhiraj Raj Gyawali and András Bárdossy

Institute for Modelling Hydraulic and Environmental Systems (IWS), University of Stuttgart, 70569 Stuttgart, Germany

**Correspondence:** Dhiraj Raj Gyawali (dhiraj.gyawali@iws.uni-stuttgart.de)

**Abstract.** Given the importance of snow on different land and atmospheric processes, accurate representation of seasonal snow evolution including distribution and melt volume, is highly imperative to any water resources development trajectories. The limitation of reliable snow-melt estimation in the mountainous regions is however, further exacerbated with data scarcity. This study attempts to develop relatively simple extended degree-day snow-models driven by freely available snow-cover images.

This approach offers relative simplicity and plausible alternative to data intensive models as well as in-situ measurements and have a wide scale applicability, allowing immediate verification with point measurements.

The methodology employs readily available MODIS composite images to calibrate the snow-melt models on spatial snow-distribution in contrast to the traditional snow-water equivalent based calibration. The spatial distribution of snow-cover is simulated using different extended degree-day models with parameters calibrated against individual MODIS snow-cover im-

ages for cloud-free days or a set of images representing a period within the snow season. The study was carried out in Baden-Württemberg in Germany, and in Switzerland. The simulated snow-cover show very good agreement with MODIS snow-cover distribution and the calibrated parameters exhibit relative stability across the time domain. Furthermore different thresholds that demarcate snow and no-snow pixels for both observed and simulated snow-cover were analysed to evaluate these thresholds' influence on the model performance and identified for the study regions.

The melt from these calibrated snow-models were used as standalone inputs to a modified HBV without the snow component in all the study catchments, to assess the performance of the melt outputs in comparison to a calibrated standard HBV model. The results show an overall increase in Nash-Sutcliffe Efficiency (NSE) performance and a reduction in uncertainty in terms of model performance. This can be attributed to the reduction in the number of parameters available for calibration in the modified HBV, and an added reliability of the snow accumulation and melt processes inherent in the MODIS calibrated snow-model

output.

This paper highlights that the calibration using readily available images used in this method allows a flexible regional calibration of snow-cover distribution in mountainous areas with reasonably accurate precipitation and temperature data and globally available inputs. Likewise, the study concludes that simpler specific alterations to processes contributing to snow-melt can contribute to reliably identify the snow-distribution and bring about improvements in hydrological simulations owing to

better representation of the snow processes in snow-dominated regimes.

# 1 Introduction

Reliable representations of spatial distribution of seasonal snow and subsequent snow-melt are critical challenges for hydrological estimations, given their crucial relevance in mountainous regimes especially because of the high sensitivity to climate change. Considering the snow effect on land and atmospheric processes, accurate representation of seasonal snow evolution is
thus highly imperative to strengthen water resources development trajectories in these regions (Kirkham et al., 2019; Schmucki et al., 2014; He et al., 2014). Various modeling and measurement techniques are currently in practice which attempt to estimate distribution of snow but these methods hold their own limitations. Prior studies on the comparison of snow models (Feng et al., 2008; Rutter et al., 2009) have highlighted the higher reliability of physically based approaches such as the energy balance approach in simulating the snow conditions. These complex models, though offer a more realistic physical detail of the sub-
processes (Wagner et al., 2009), are often associated with intensive data requirement, which is generally a big limitation in mountainous catchments around the world (Girons Lopez et al., 2020). Likewise, in-situ measurements of snow-depth providing accurate measure, seldom cover a wider spatial extent and are prone to be non-representative due to local influences. Lack of snow-depth information and to some extent, persistent cloud cover in the mountains limit the standalone usage of Remote sensing images in snow estimation (Tran et al., 2019). However, these images can provide a plausible alternative to ground
based data especially in the data scarce mountainous regions, since their resolution and availability do not depend on the terrain (Parajka and Blöschl, 2008).

The MODerate resolution Image Spectralradiometer (MODIS) (Hall et al., 2006) on board Terra and Aqua satellites provide one of the most extensively used snow-cover products worldwide owing to their daily temporal resolution and a high spatial resolution of 500m at the Equator. The MODIS snow-cover data performance, though seasonally and region dependent, has
been found to be accurate enough for hydrological context (Parajka and Blöschl, 2008). The cloud obstruction in MODIS, though significant, can be reduced combining the Aqua and Terra MODIS images and other spatio-temporal filtering techniques (Tran et al., 2019; Gafurov and Bárdossy, 2009; Wang and Xie, 2009).

Remote sensing integration in hydrological modeling has gained important strides in the recent years (Wagner et al., 2009). More so, several studies have been carried out coupling satellite-based snow cover information with hydrological models such
as ASCAT soil water index, MODIS snow-cover and discharge based calibration of a hydrological model (Tong et al., 2021); MODIS snow-cover constrained multi-variable discharge calibration for hydrological models (Parajka and Blöschl, 2008; Udnæs et al., 2007); MODIS snow-covered area based calibration of a distributed hydrological model (Franz and Karsten, 2013); model initialization (Liu et al., 2012). These studies highlight the value of MODIS snow-cover information in multi-variable calibration in addition to discharge, leading to a better representation of the hydrological processes, reduced uncertainty, and to
some extent improved hydrological predictions. Apart from the multi-calibration approaches, Széles et al. (2020) implemented a step-based calibration technique to calibrate the individual modules of a hydrological model including snow, soil moisture and runoff generation processes, which was concluded to be a well-informed runoff simulation. They used MODIS snow-cover data as a gap-filling information for the missing observed time-lapse photos of snow-cover. Tekeli et al. (2005) used MODIS products in identifying the snow duration curve to be used in a snow-melt model and concluded that the coupling provides

crucial information on snow-melt timing and magnitude. Likewise, there have been several studies to compare and improve the snow routine in various hydrological models in recent years. Girons Lopez et al. (2020) evaluated various formulations of the temperature-index approach to analyze their response via the HBV model in 54 mountainous European catchments. Caicedo et al. (2012) also identified the best performing variants of degree-day calculations for different regions in Colombia. They concluded that these specific targeted alterations improve the performance in terms of snow processes.

This paper draws its motivation from these studies to improve the simulation of snow-accumulation and melt processes in data-scarce mountainous regions, using freely available data like MODIS snow-cover products. The widely used point-based snow depth and snow water equivalent measurements are often not representative due to the high variability of snow accumulations rendering calibration of snow-melt models based on point measurements highly uncertain. A standalone calibration of snow-melt modules based solely on pixel-wise MODIS snow-cover information was not found to be widely implemented. Franz and Karsten (2013) tested a MODIS fractional snow-cover area (SCA) based calibration of a distributed hydrological model SNOW17 with areal snow depletion concept in which they concluded that calibrating only on MODIS SCA did not bring any improvement in the hydrological predictions. However, we identified the need to further explore the gap in exploiting the spatial distribution information of snow, which MODIS snow-cover can provide on a daily-basis and a reasonable spatial detail.

The purpose of this paper is, thus, to present a simple calibration method for snow accumulation and melt based on the satellite-based spatial binary ('snow', 'no-snow') snow cover information. This offers a wide scale applicability, allows immediate verification with point measurements, and holds a high relevance in data scarce regions, particularly in identifying time-continuous snow extents (with depth information) free from highly localized influences. The novelty of this study is to propose a standalone approach using MODIS snow-cover images for calibration of independent conceptual snow-melt models, thereby estimating model parameters from individual or sets of MODIS images. The data and the methodological approach can calibrate relatively complex snow-melt modules with reasonably accurate precipitation and temperature data without over-calibration, mainly owing to the robust binary data selected for calibration and the spatial extent of the satellite images. This also allows for the formulation of a flexible snow-melt module useful for distributed hydrologic modeling.

With the degree-day approach as a basis, we implemented different modifications to the snow-melt models incorporating different aspects governing snow hydrology such as precipitation-induced melt, radiation, and topography. For a better representation of the model performances, we evaluated the models in different layers of added detail. Sensitivity analyses for different thresholds that demarcate snow and no-snow pixels for both observed and simulated snow-cover, in the study domain were carried out to evaluate these thresholds' influence on the model performances. To better understand and to validate the efficacy of the approach in terms of discharge, the calibrated melt outputs from the snow-melt model were also evaluated in a discharge-calibrated hydrological model. This allowed for the identification of more robust parameter sets as the model uncertainty related to snow processes is significantly reduced. A simpler calibration of the hydrological models was achieved, as there were less parameters to be estimated. This approach also tends to reduce the hydrological model uncertainty as the set of equifinal parameters becomes smaller, which is a known challenge in hydrological modeling (Beven, 2001).

## 2 Study Area and Data

To develop and test simple snow models in the mountainous / snow dominated regions, the study area is selected as two distinct snow-regimes, a) characterized by intermittent snow and b) characterized by partly longer duration snow. For the former, Baden-Württemberg (BW) region in Germany was selected. Whole of Switzerland was considered to represent the longer duration snow for the study. Figure 1 shows the study domain:

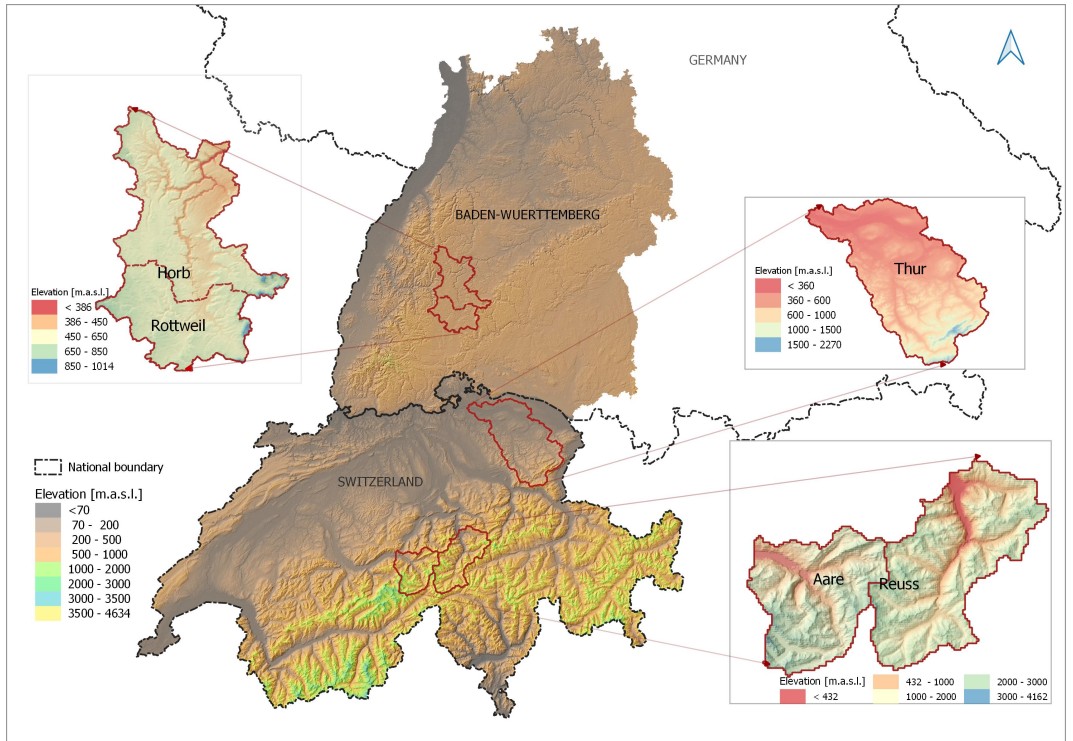

**Figure 1.** Map of the study area with elevation

The BW region includes the Schwabian Alps with the elevation rising to 1465 m.a.s.l. from a lowest of 88 m.a.s.l. Likewise,
Switzerland includes the Swiss Alps region which covers the perennial snow/glacier area. The elevation ranges from below 200 to 4634 m.a.s.l.. The study areas exhibit an average snow season from October to April in Germany and September to June in Switzerland. For hydrological modeling, five catchments, viz, Neckar catchment at Rottweil and Horb in BW, and Reuss catchment at Seedorf, Aare catchment at Brienzwiler and Thur catchment at Andelfingen in Switzerland, were selected. The Reuss and Aare catchments have a longer snow-cover around the year and include glaciated areas. The properties of the
catchments are shown in Table 1.

The data used for the study are:

**Table 1.** Catchment properties

| River | Outlet | Catchment Area, $km^2$ | Catchment Elevation, m.a.s.l. | | | Glaciation,% |
|---|---|---|---|---|---|---|
| | | | Max | Min | Mean | |
| Neckar | Rottweil | 412 | 1006 | 555 | 705 | 0 |
| Neckar | Horb | 1110 | 1006 | 386 | 656 | 0 |
| Reuss | Seedorf | 837 | 3416 | 437 | 2010 | 6.4 |
| Thur | Andelfingen | 1702 | 2217 | 372 | 770 | 0 |
| Aare | Brienzwiler | 555 | 3798 | 580 | 2135 | 15.5 |

- **Hydro-meteorology** : Daily station meteorological data viz. precipitation, and minimum, maximum and mean temperatures from 2010-2018 were acquired for the study. For Germany, these variables were obtained from the Deutsche Wetterdienst (DWD), and from Federal Office of Meteorology and Climatology (MeteoSwiss) for Switzerland. Likewise daily discharge timeseries for selected catchments were acquired from Bundesanstalt für Gewässerkunde (BFG) for Germany and Federal Office for the Environment (FOEN) for Switzerland.

- **Topography**: Shuttle Radiation Topography Mission (SRTM) 90m resolution Digital Elevation Model (DEM) (Jarvis et al., 2008) was used in the study. The DEM was rescaled to match the MODIS resolution for consistency. Likewise, aspect and slope rasters were also obtained from this DEM.

- **Snow-cover**: Daily MODIS Terra and Aqua snow-cover data Version 6 (Hall and Riggs, 2016) from 2010 to 2018 were used for calibrating the models and further analysis of snow distribution in the study regions. The resolution of the data is 500m at the Equator.

This study presents a distributed modeling approach with model computations done at pixel level of a gridded domain of 464m x 464m grids. For this, the input data were pre-processed and interpolated onto the aforementioned grid cells. A gridded schema was extracted for both regions using the MODIS snow-cover data as a reference. This schema was considered as the reference gridded domain for the data interpolation and model run.

## 2.1 MODIS pre-processing and cloud removal

The Aqua and Terra variants of MODIS snow-cover data were downloaded and then pre-processed. The pre-processed images were sequentially and spatio-temporally filtered using a cloud-removal procedure as described in Gafurov and Bárdossy (2009). The procedure follows the steps below:

(a) The first step checks the Aqua-Terra image combination. Pixels with clouds (255) in one of the images and land (0) or snow (1-100) in the other was replaced with the snow / land value and vice versa. The output is a combined raster with reduced cloud pixels. The combined raster was then reclassified as ‚0' and ‚1'. No snow pixel values of the combined raster (0) are set to ‚0' and snow pixel values (1-100) are set to ‚1'. Everything else is set to 'No data'.

(b) The second step compares the preceding and succeeding days for a pixel under consideration. If both the days for the pixels are cloud-free with 0 or 1, the pixel under consideration will respectively get either 0 or 1 for the day.

(c) Likewise the third step compares two days backward and one day forward, and one day backward and two days forward combination to check for the cloud free days and infill accordingly, assuming consecutive snow or no-snow days.

(d) The fourth step compares the lowest elevation with snow and the highest elevation without snow for each day. Any pixel with elevation higher than the lowest elevation snow pixel would get ‚1‘ and the elevation lower than the highest elevation without snow would get ‚0‘.

(e) The fifth step searches for '0' or '1' in a 8 pixel neighbourhood surrounding the cell. If the neighbourhood has a mode at least 4 valid values, the pixel will then be either '0' or '1'.

## 2.2 Spatial interpolation of precipitation and temperature

Both BW and Switzerland have a well distributed and dense network of meteorological stations. The daily precipitation and temperature values from these stations were used for geostatistical interpolation onto the aforementioned schema for the regions.

For the interpolation of temperature data, External Drift Kriging (EKD) was opted in the regions under study, with station elevation as a drift (Hudson and Wackernagel, 1994). The station elevation exhibits strong correlation with the monthly and seasonal temperatures. Daily minimum, maximum and mean temperatures from 85 stations in Baden-Württemberg and 365 stations for Switzerland were used for the interpolation. Cross validation using leave-one-out approach was carried out to check for the applicability and the quality of the EKD interpolation.

For precipitation,the daily precipitation sums were interpolated onto the schema using a detrended Residual Kriging (RK) (Phillips et al., 1992; Martínez-Cob, 1996). To improve the precipitation interpolation in the higher elevation, a multiple linear regression (MLR) approach using directionally smoothed elevation was carried out for the study. Directional smoothing of elevation was done using half-space smoothing (Bárdossy and Pegram, 2013). The approach uses a directionally transformed and smoothed topography to identify the effect of directional advection for each day. Eight different directions with 45 degrees incremental angles, and 3 different smoothing distances (2, 3 and 5 kms) were considered in this study. For each time-step, a simple optimization was done to assess the correlation of the precipitation with the shifted DEMs, and the best direction and the smoothing radius for the timestep were identified. This shifted and smoothed elevation was then used along with X and Y coordinates of the stations in the MLR to obtain precipitation estimates for stations. The residuals were then calculated for each day and ordinary Kriging was carried out to obtain the Kriged residuals. MLR estimated precipitation surfaces for each time step using X and Y coordinates and shifted elevation for the grid points were then added to the Kriged residual surfaces to obtain the final precipitation estimates. 224 stations in BW, and 449 stations in Switzerland were used in this study. Leave-one-out cross validation for each station was done for both variables.

## 3 Methodology

The methodological framework applied for the study is shown in fig. 2 and is further discussed in subsequent sections.

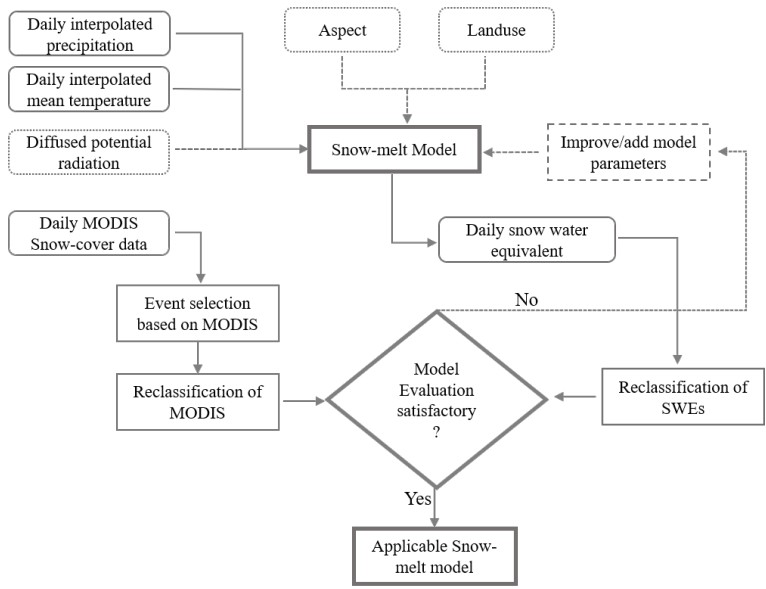

**Figure 2.** Methodological approach for the study

### 3.1 Model Variants

This study employs empirical, temperature-index melt modelling approach using the degree day factors. The degree day models are widely used owing to relatively easier interpolation of air temperature, and reasonable computational simplicity (Hock, 2003). This degree-day approach assumes melt rate as a linear function of the air temperature. Due to inherent large-scale spatial variability in the mountain regions, distributed meteorological inputs, were employed to drive the different variants of the extended degree-day snow-melt models on a daily timescale. The major parameters used in the models are defined below.

Where,

$P(t,x)$= precipitation amount at location x at time t, mm

$S(t,x)$= snow water equivalent amount at location x at time t, mm

$T_{av}(t,x)$= mean temperature at location x at time t, °C

$P_s(t,x)$ = water equivalent of precipitation falling as snow at location x at time t, mm

$M_s(t,x)$ = melt water amount at location x at time t, mm

$T_T$ = threshold critical temperature defining snow or no snow, °C

$D_s$ = dry degree day factor, $mm°C^{-1}$

$T_{mx}(t,x)$= maximum temperature at location x at time t, °C

$T_{mn}(t,x)=$ minimum temperature at location x at time t, °C

$scf=$ snow correction factor to account for the gauge undercatch of snow

The following model variants were used to estimate the snow-water equivalent (SWE, mm) and the resulting snow-cover in each pixel. Different nomenclatures are given to the models for the ease of understanding. Each successive model represents a gradual parameter wise modification to the basic degree-day model.

*Basic Degree-day Model (Model 1)*

Model 1 is the most basic of all model variants. This model estimates the melt for each time-step as a linear function of 185 the difference between daily mean temperatures and a threshold temperature value demarcating liquid precipitation and snow precipitation. A degree day factor controls the rate of melt. Eq.(1) calculates the amount of SWE available in pixel 'x' at time 't'. Similarly the snow-precipitation and the resulting melt are calculated with Eqs. 2a and 2b as the model basis for each pixel, 'x' in the study domain. A correction factor to account for the snowfall undercatch by the gauges and the vegetation interception $scf$ is also used in this model and extended to all models in the study.

$$S(t,x) = S(t-1,x) + P_s(t,x) - M_s(t,x),  \tag{1}$$

Where,

$$P_s(t,x) = \begin{cases} P(t,x) \cdot scf & \text{if } T_{av}(t,x) < T_T \\ 0 & \text{if } T_{av}(t,x) \geq T_T \end{cases} \tag{2a}$$

$$M_s(t,x) = \begin{cases} 0 & \text{if } T_{av}(t,x) < T_T \\ min(S(t,x), \ D_s\ (T_{av}(t,x) - T_T) & \text{if } T_{av}(t,x) \geq T_T \end{cases} \tag{2b}$$

*Wet Degree-day Model (Model 2)*

To account for the melt induced by rain at temperatures higher than the critical threshold temperature, this variant adds a precipitation melt factor which controls the rate of melt based on air temperature and the precipitation amount falling on the pack. Similar approach was discussed in Bárdossy et al. (2020). This melt factor, henceforth referred to as $D_w$ increases the 200 melt from Eq.(2b) on days with precipitation higher than a threshold value. For a given wet day i.e., $P(t,x) > P_T$, the melt is calculated as in Eq.(3). For a dry day, melt is calculated as Eq.(2b).

$$M_s(t,x) = \begin{cases} 0 & \text{if } T_{av}(t,x) < T_T \\ min(S(t,x), \ D(t,x)\ (T_{av}(t,x) - T_T) & \text{if } T_{av}(t,x) \geq T_T \end{cases} \tag{3}$$

Where,

$$D(t,x) = D_s + D_w(P(t,x) \text{ - } P_T)$$

$P_T$ = Threshold precipitation depth beyond which the liquid precipitation contributes to melt, mm

$D_w$ = the wet melt factor, $mm.mm°C^{-1}$

$D(t,x)$ = combined melt factor on wet days, $mm°C^{-1}$

### Wet Degree-day Model with snowfall and snow-melt temperatures (Model 3)

The instantaneous forms of precipitation as snow and liquid gives a clear indication of two temperature thresholds which demarcate the solid and liquid state of precipitation (Schaefli et al., 2005). This model includes different snowfall and snow-melt temperatures in Model 2 for a more accurate representation of the liquid to snow phase partition and melt initiation. This has been previously discussed in (Debele et al., 2009; Girons Lopez et al., 2020). For temperatures in between, snow is linearly interpolated for the day as a proportion of the precipitation. The formulation of the model are given by Eqs.(4) and (5).

$$P_s(t,x) = \begin{cases} P(t,x) & \text{if} \quad T_{av}(t,x) < T_S \\ P(t,x) \cdot \left( \frac{T_{av}(t,x) - T_M}{T_S - T_{av}(t,x)} \right) & \text{if} \quad T_S \leq T_{av}(t,x) \leq T_M \\ 0 & \text{if} \quad T_{av}(t,x) > T_M \end{cases} \tag{4}$$

$$M_s(t,x) = \begin{cases} 0 & \text{if} \quad T_{av}(t,x) < T_M \\ min(S(t,x), \quad D(t,x) \ (T_{av}(t,x) - T_M)) & \text{if} \quad T_{av}(t,x) \geq T_M \end{cases} \tag{5}$$

Where,

$T_S$ and $T_M$ are the snowfall and snow-melt temperatures respectively.

### Aspect distributed snowfall temperatures (Model 4)

This model was envisioned with an assumption that topographical aspect plays a major part in the spatial distribution of snow-fall and snow-melt temperatures. Based on this assumption, this variant distributes the snowfall temperature in Model 3, according to the topographical aspect. The snowfall temperature distribution is done by Eq.(6) :

$$T_{S,x} = T_{Smin} + (T_{Smax} - T_{Smin}) * [0.5 * \cos(aspect_x) + 1]^{PF} \tag{6}$$

Where,

$T_{Smin}$ = lower bound of the snowfall temperature

$T_{Smax}$ = upper bound of the snowfall temperature

$aspect_x$ = topographical aspect of grid 'x', (radians)

$PF$ = power factor to distribute the aspect

### *Aspect distributed snow-melt temperatures (Model 5)*

In general, south facing slopes are warmer in the Northern hemisphere resulting in a faster melt of snow compared to the north facing slopes. This model, thus, distributes the snow-melt temperature in Model 3 within a range defined by minimum and maximum bounds of snow-melt temperature, according to the topographical aspect. The snow-melt distribution is represented
by Eq.(7) :

$$T_{M,x} = T_{Mmin} + (T_{Mmax} - T_{Mmin}) * [0.5 * \cos(aspect_x) + 1]^{PF} \tag{7}$$

Where,

$T_{Mmin}$ = lower bound of the snowfall temperature

$T_{Mmax}$ = upper bound of the snowfall temperature

$aspect_x$ = topographical aspect of grid 'x', (radians)

$PF$ = power factor to distribute the aspect

### *Radiation Induced melt Model (Model 6)*

The integration of radiation information in degree-day models can lead to better estimation of snow-melt (Hock, 2003). This
model was formulated to accommodate the radiation data in addition to the aspect-based temperature distribution. The radiation induced melt was added to Model 5 by incorporating the diffused incident radiation on the snow pixel on a cloud-free day. The incident global radiation is calculated using a viewshed based algorithm "r.sun algorithm" (Hofierka and Suri, 2002; Neteler and Mitasova, 2002) and has an added advantage of radiation distribution in the valleys. Daily temperature difference (Tmax -Tmin) for each grids was also calculated using interpolated daily minimum and maximum temperatures and was used as a
cloud cover proxy. For this study, pixels with a daily temperature difference above a certain threshold were assumed to be cloud free and this is where radiation induced melt becomes active. Likewise, temperature differences lesser than the threshold render the pixels cloudy. The diffusion factor ranging from 0.2 for clear sky conditions to 0.8 for overcast conditions diffuses the incoming radiation. The radiation induced melt is added to the melt outputs from the preceding models on cloud-free pixels and is calculated using Eq.(8). Figure 3 shows an example of diffused radiation calculated for a cloud free day in
Baden-Württemberg.

$$M_{s-R}(t,x) = \begin{cases} (1-alb) \cdot r_{ind} \cdot R_D(t,x) & \text{if } T_{mx}(t,x) - T_{mn}(t,x) \geq 5°C \\ 0 & \text{if } T_{mx}(t,x) - T_{mn}(t,x) < 5°C \end{cases} \tag{8}$$

Where,

$M_{s-R}(t,x)$ = Radiation induced melt at grid x at time t, mm

$R_D(t,x)$ = Diffused radiation at grid x at time t, $Wh \cdot m^{-2}day^{-1}$

$alb$ = Albedo of snow

$r_{ind}$ = Radiation melt factor, $mm \cdot (Wh.m^{-2}day^{-1})$

$(T_{mx}(t,x) - T_{mn}(t,x))$ = temperature difference at time t, as a cloud proxy to define clear-sky and overcast conditions

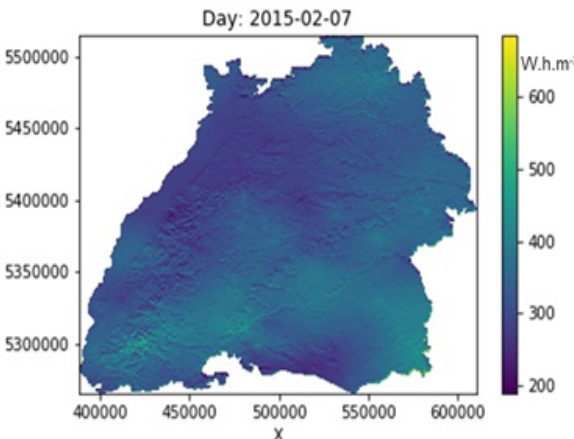

**Figure 3.** Illustration of diffused radiation calculated using r.sun.daily algorithm for Baden-Württemberg

### 3.2 Data requirement of the models

Table 2 summarizes the input data requirement for each model. The major inputs are the DEM, precipitatiton and temperature. The other variables are the derivatives from these major inputs. For instance, daily temperature difference was considered as a proxy for the cloud information. The aspect information and daily global radiation are derived from the DEMs. In addition to the data presented in the table, the daily MODIS snow-cover distribution is also required for model calibration and evaluation. Freely available inputs such as the DEM and the MODIS images provide a crucial flexibility with minimum data requirement to drive the snow-melt models. Likewise, daily observed stream-flows are also used for calibration and validation of the HBV model.

### 3.3 Model calibration

The model calibration in this study was done in two distinct steps. Firstly, the six aforementioned snow-melt model variants were calibrated using snow-cover distributions. The second calibration step was done for two hydrological models, namely

**Table 2.** Inputs required for the different model variants

| Models | Spatial inputs | Spatio-temporal inputs (daily) | | |
|---|---|---|---|---|
| | DEM | Precipitaion, mm | Mean Temp.,°C | Max. / Min Temp, °C |
| Model 1 | yes | yes | yes | - |
| Model 2 | yes | yes | yes | - |
| Model 3 | yes | yes | yes | - |
| Model 4 | yes | yes | yes | - |
| Model 5 | yes | yes | yes | - |
| Model 6 | yes | yes | yes | yes |

the original HBV (Bergström, 1995), henceforth termed as 'standard HBV' and a HBV model modified to accommodate the standalone melt outputs from the best performing snow-routine variant as inputs, from hereon termed as the 'modified HBV'.

In order to calibrate the snow-model parameters, the snow-cover simulated by the snow model was compared with the MODIS observations on selected days or periods with available data. A pixel was considered as snow-covered if the simulated snow water equivalent exceeded a pre-considered snow water equivalent threshold of 0.5 mm which corresponds to a snow depth of approximately 2.5 mm.

The calibration is based on the Brier-score (BS) (Eq.(9)). It is a score function that measures the accuracy of probabilistic predictions. The BS in this study refers to the mean squared error between observed binary patterns of snow/no snow from MODIS and the ones simulated by the extended degree-day models. The Brier-score varies between 0 and 1 with the values closer to ‚0' indicating better agreement between the model outputs and the MODIS image.

$$BS(t) = \frac{1}{N} \sum_{t=1}^{N} (f_i(t) - O_i(t))^2 \tag{9}$$

Where,

$f_i(t)$ = simulated snow-cover (0/1) on day $t$ and pixel $i$,     $o_i(t)$ = observed snow-cover (0/1) on day $t$ and pixel $i$

The objective function is the sum of the BS values over the days with observed MODIS snow-cover:

$$OF = \sum_{k=1}^{K} BS(t_k) \tag{10}$$

Where,

$t_k$ are the days with observed MODIS snow-cover,

The snow model parameters were identified by minimizing objective function in Eq.(10). In order to reflect the equifinality imparted by the model , the Robust Parameter Estimation (ROPE) methodology was applied for the model parameter optimization. ROPE uses the concept of data depths to identify best-performing robust parameter sets and their properties for different

calibration periods in different catchments, with an underlying assumption that it identifies parameters sets without overemphasizing the processes defined by the parameters. Further details regarding ROPE can be found in Bárdossy and Singh (2008). For the calibration, the following steps were carried out.

1. The bounds for the model parameters were set.

2. 'N' random parameter sets ($X_N$) were generated in the 'd' dimensions space limited by the bounds set in Step 1.

3. The models (snow-melt, standard HBV and the modified HBV) were run for each parameter sets and the corresponding objective functions were calculated.

4. Based on the model performance, a pre-defined subset of the best performing parameter sets $X_N^{'}$ were drawn.

5. 2*N random parameter sets were again generated within the bounds of the best performing sets from Step 3.

6. A set of $Y_M$ parameters were identified where for each vector $\theta \in Y_M$, the depth calculated with respect to the subset $X_N^{'}$ is greater than 0, i.e $D(\theta) > 0$.

7. Within the bounds defined by the 'deep' parameter sets in Step 6, further 'N' parameter sets were generated so that $X_N$ = $Y_M$.

8. Steps 3-6 were repeated for various iterations assuming that the performance corresponding to $Y_M$ does not differ more than what one would expect from the observation errors.

A set of 1000 heterogeneous parameter vectors with similar model performance in terms of the objective function were generated. These sets of 'good' points can be defined as the parameter sets that are less-sensitive and transferable, thereby providing a 'compromised' solution. These parameter vectors were estimated for each region assuming spatio-temporally constant/variable (wherever possible) parameter distribution and were estimated within a plausible range as described in different snow modelling studies.

ROPE was applied to calibrate the snow-melt models, the standard HBV model with snow, and the modified HBV with external melt for this study. For the snow-melt models, the calibration was initially done for both regions on daily snow-cover images with more than 60% valid pixels (< 40% cloud cover) for the snow season in different years. The snow season was selected as October - May for Baden-Württemberg and September - June for Switzerland, assuming a possible snow-cover being present for the time period.

The second calibration was the hydrological calibration with discharge data at the catchment level and was done for the whole years of 2011-2015 in Baden-Württemberg and 2011-2018 in Swiss catchments with 2010 as the warm-up year. Here the modified HBV had the melt from the selected snow-melt model as standalone input. The discharges simulated by the standard HBV and the modified HBV were compared for each of the catchments. Nash-Sutcliffe Efficiency (NSE, Eq.(11)) was used to evaluate the performance of the melt inputs, where the simulated and observed variables refer to modelled and observed discharge at time $t$.

$$NSE = 1 - \frac{\sum_{t=1}^{T} \left( Y_o^t - Y_m^t \right)^2}{\sum_{t=1}^{T} \left( Y_o^t - \bar{Y}_o^T \right)^2} \tag{11}$$

Where,

$Y_m^t$ = Simulated variable at time t,

$Y_o^t$ = Observed variable at time t,

$\bar{Y}_o^T$ = mean of observed variable for the time period T,

$T$ = length of time series,

## 3.4 Model validation

The calibrated parameter vectors were used to validate the simulated snow-patterns for different seasons using the sets of MODIS images representative of the season as well as on individual images representing unique isolated events, for both BW and Switzerland. To analyse the performance of the snow models at a catchment level and subsequently for the discharge evaluation, five catchments viz. Neckar-Horb and Neckar-Rottweil in BW, and Reuss-Seedorf, Thur-Andelfingen and Aare-Brienzwiler in Switzerland, were selected. At the catchment level, the snow routine parameters from the 1000 best parameter sets of the standard HBV model calibrated on discharge were subset and used to simulate the snow-distribution in all the catchments for the same time period. The corresponding Brier-scores were calculated. The validation was then done as a comparison of the Brier-scores calculated from the 1000 best parameters for the selected snow-melt model calibrated on MODIS images (for each catchment) and the 1000 best Brier-scores obtained from the snow-routine of the standard HBV model calibrated on discharge.

To validate the the performance of melt outputs from the snow-cover calibrated snow-melt models, the 1000 best NSEs from the standard HBV model and the modified HBV model, both calibrated on discharge for the same calibration period for all 5 catchments were compared. The evaluation was assessed based on the ranges and dispersion of the two sets of best NSEs.

## 3.5 Model uncertainty

A common problem of hydrological modelling is that due to the inaccurate observations and simplified representation of the relevant hydrological processes, the parameters of the models cannot be identified accurately. Due to this equifinality, a set of acceptable model parameters can be assessed. If one can use specific additional information, the set of acceptable model parameters may reduce, leading to uncertainty reduction. In the case of snow modelling, the parameters of a hydrological model can be split into two distinct parts, the snow accumulation and melt parameters $\theta_s$ and the other model parameters $\theta_m$. If the snow model parameters $\theta_s$ are calibrated independently of the model parameters, $\theta_m$, one receives a parameter set $M_s$ - as the same problem of equifinality occurs for the snow models too. For each $\theta_s \in M_s$, one can calibrate the hydrological model parameters $\theta_m$ leading to a set $M_m$. This way a well performing parameter set $M_{sm} = M_s \times M_m$ can be obtained. However if the parameters of the model $\theta_m$ such that the model quality is the same as for calibrating the parameters $(\theta_s, \theta_m)$

jointly (without using snow observations) obtaining the parameter set $M$, then the parameter set $M_{sm} = M_s \times M_m \subset M$. This is because all parameter combinations in $M_{sm}$ could also be obtained from the traditional model calibration, but there are parameters in $M$ where parameter compensations lead to snow parameters $\theta_s$ which are not acceptable for the snow model evaluation. Thus model calibration of conceptual model may not lead to a better model performance, but instead can reduce uncertainty. On the other hand, the separate calibration of the hydrological model and the snow-melt model makes it possible to include more parameters into the snow model. If the same model would be calibrated together with the hydroligcal model, the increase of the number of parameters would lead to a much more complex calibration procedure. In the results section this is demonstrated for the models considered.

## 4   Results

### 4.1   Model Results

**Switzerland**

For the calibration of the snow-melt model variants, a set of MODIS images for the whole snow season of 2012-09-01 till 2013-06-30 for Switzerland was selected as the reference snow-cover distribution series. The calibration performance of model variants are expressed in terms of Brier scores as well as confusion matrices depicting the proportion of true and false identifications of snow and no-snow pixels. The overall model error, i.e the sum of falsely identified instances of snow and no-snow pixels is the Brier-score.

Table 3a shows the calibration results of different model variants as normalized confusion statistics calculated for the reference time period for Switzerland. The columns of the confusion statistics table indicate the proportions of true negatives (both 'no snow'), false positives (MODIS: 'no snow', simulated: 'snow'), true positives (both 'snow') and false negatives (MODIS: 'snow', simulated: 'no snow'). All the six models reported good Brier-scores ranging from 0.084 to 0.095. The results indicate that the models have very close performance when calibrated over the course of a whole snow season, in Switzerland. The Model 6, with the radiation induced melt has the best performance in terms of overall Brier-scores as well as the reduction in the false recognition of snow. This model, however, slightly overestimates the snow in the region.

Figure 4a shows the validation of the radiation-based model on the snow-cover image for a relatively cloud-free day (2013-01-08). The figure shows that the model calibrated on the whole season adeptly mimics the MODIS snow-cover distribution for the day with a very good Brier-score of 0.077. The left plot in the figure is the MODIS image for the reference day, the central plot shows the simulated image for the day and the right one shows the differences in prediction.

The model performances were further scrutinized in different elevation zones in both regions in terms of under-, over- and total estimation errors. Under-estimation error is the average normalized false negative instances, over-estimation error is the average normalized false positive instances and total error is the mean Brier-scores for each elevation zones. Figure 5a shows the results in Switzerland. Model 6 showed a reduction in over-estimation error throughout the elevation zones in comparison to other models. However, the model was underestimating snow for elevations below 1500 m.a.s.l. The overall

**Table 3.** Normalized confusion matrices for the calibration periods (a) Switzerland for 2012-09-01 till 2013-06-30 (b) Baden-Württemberg for 2012-10-01 till 2013-05-31

(a) Switzerland

|  | True positive | False positive | True negative | False negative | Brier-score |
|---|---|---|---|---|---|
| Model 1 | 0.625 | 0.036 | 0.280 | 0.059 | 0.095 |
| Model 2 | 0.621 | 0.039 | 0.289 | 0.050 | 0.089 |
| Model 3 | 0.626 | 0.035 | 0.289 | 0.051 | 0.086 |
| Model 4 | 0.625 | 0.035 | 0.289 | 0.050 | 0.085 |
| Model 5 | 0.622 | 0.039 | 0.293 | 0.047 | 0.086 |
| Model 6 | 0.610 | 0.050 | 0.306 | 0.034 | 0.084 |

(b) Baden-Württemberg

|  | True positive | False positive | True negative | False negative | Brier-score |
|---|---|---|---|---|---|
| Model 1 | 0.758 | 0.031 | 0.169 | 0.042 | 0.073 |
| Model 2 | 0.759 | 0.032 | 0.168 | 0.041 | 0.073 |
| Model 3 | 0.760 | 0.034 | 0.167 | 0.040 | 0.074 |
| Model 4 | 0.758 | 0.032 | 0.168 | 0.041 | 0.073 |
| Model 5 | 0.756 | 0.030 | 0.171 | 0.043 | 0.073 |
| Model 6 | 0.762 | 0.027 | 0.173 | 0.037 | 0.064 |

error however, remains lower for the <500 m.a.s.l and >2500 m.a.s.l regions. The mean error remained the lowest, implying the best performance among the variants.

**Baden-Württemberg**

The different snow models were also tested in this region in Germany. The region was selected to test the efficacy of the approach in shorter duration snow region. Here, the snow season of 2012-10-01 till 2013-05-31 was selected as the reference period and the corresponding MODIS images were used for calibration . As in Switzerland, all models were able to mimic the snow-distribution pattern for the reference day very well. Model 6 outperformed all the other model variants as shown by the proportion of true and false predictions in Table 3b. A gradual improvement in model performance with additional parameterization can be inferred from the table. Though comparable, Model 6 Brier scores have a starker contrast with other model results, in comparison to the Swiss results. A good improvement in true recognition and a subsequent reduction in false identification can be observed with the radiation-based model. An illustration of the validation of Model 6 on a cloud free day of 2013-03-04 can be seen in fig. 4b. The simulated image matches the MODIS image for the day with about 93% accuracy which reflects the strength of the seasonal calibration.

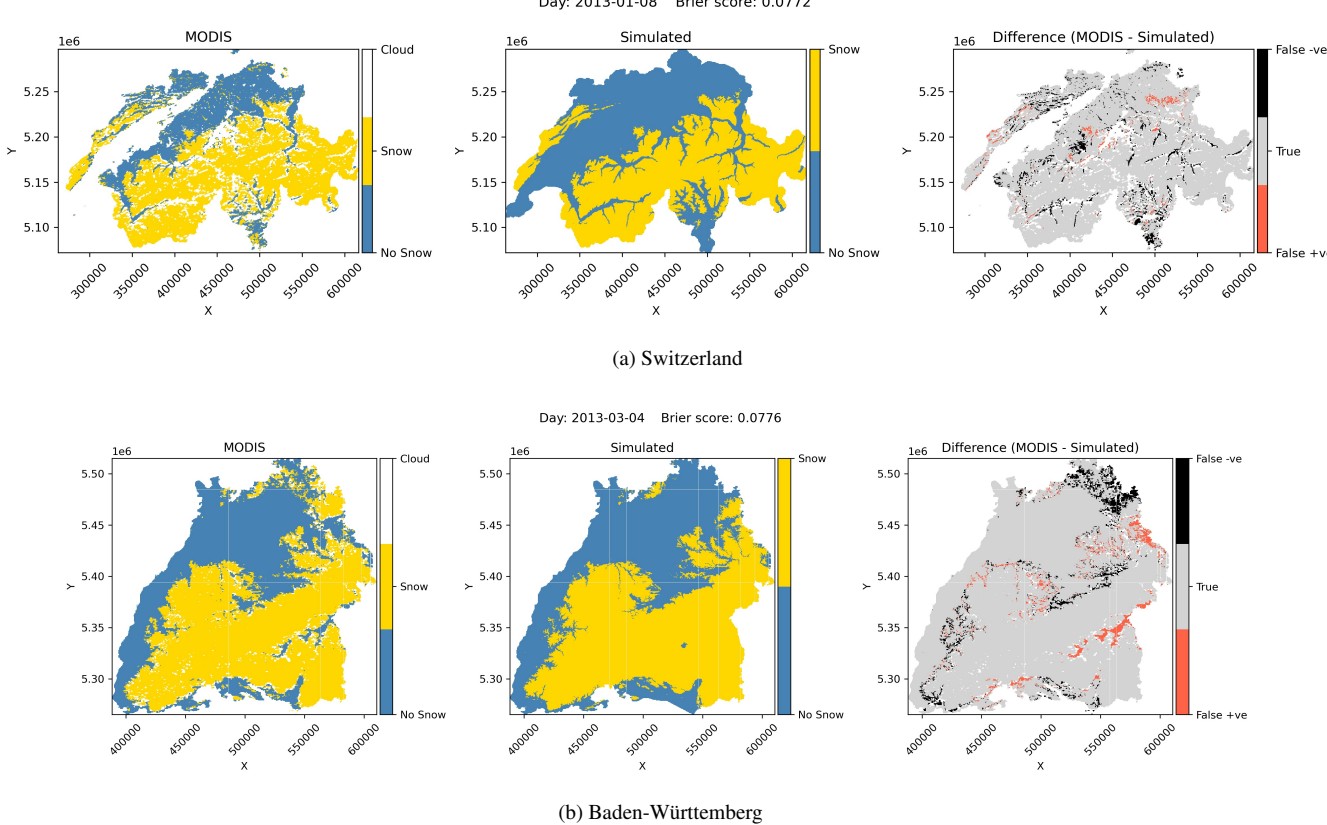

**Figure 4.** MODIS inferred (left) vs Model 6 simulated snow distribution (centre) and differences between MODIS and simulated image (right); (a) Switzerland (b) Baden-Württemberg

Likewise, elevation-based discretization of the model performance in Baden-Württemberg, as depicted by fig. 5b shows that the radiation-based model and Model 1 have the lowest under-estimation error. The error however, increases with increasing elevation. The over-estimation error is the lowest for Model 4, the trend for which decreases as the elevation increases for all but Model 1. The overall error is highly reduced with Model 6 throughout all elevation zones as indicated by the right subplot.

Though all the models perform very well in the overall scenario as well as in all the elevation zones as reflected in the performance scores, Model 6 was selected for both Switzerland and Baden-Württemberg owing to the lowest overall error and from hereon used as the reference model for further analysis.

### 4.2 Sensitivity analysis of different thresholds for 'snow/no snow' differentiation

Sensitivity analysis was carried out for different thresholds using the reference model for both Baden-Württemberg and Switzerland to identify the best snow/no snow differentiation. The time period used for the analysis was from 2013-2015. The thresholds analyzed were the Normalized Difference Snow Index (NDSI) thresholds to demarcate the snow and no snow pixels in the reference MODIS images; cloud threshold as a percentage of valid pixels to demarcate the number of daily im-

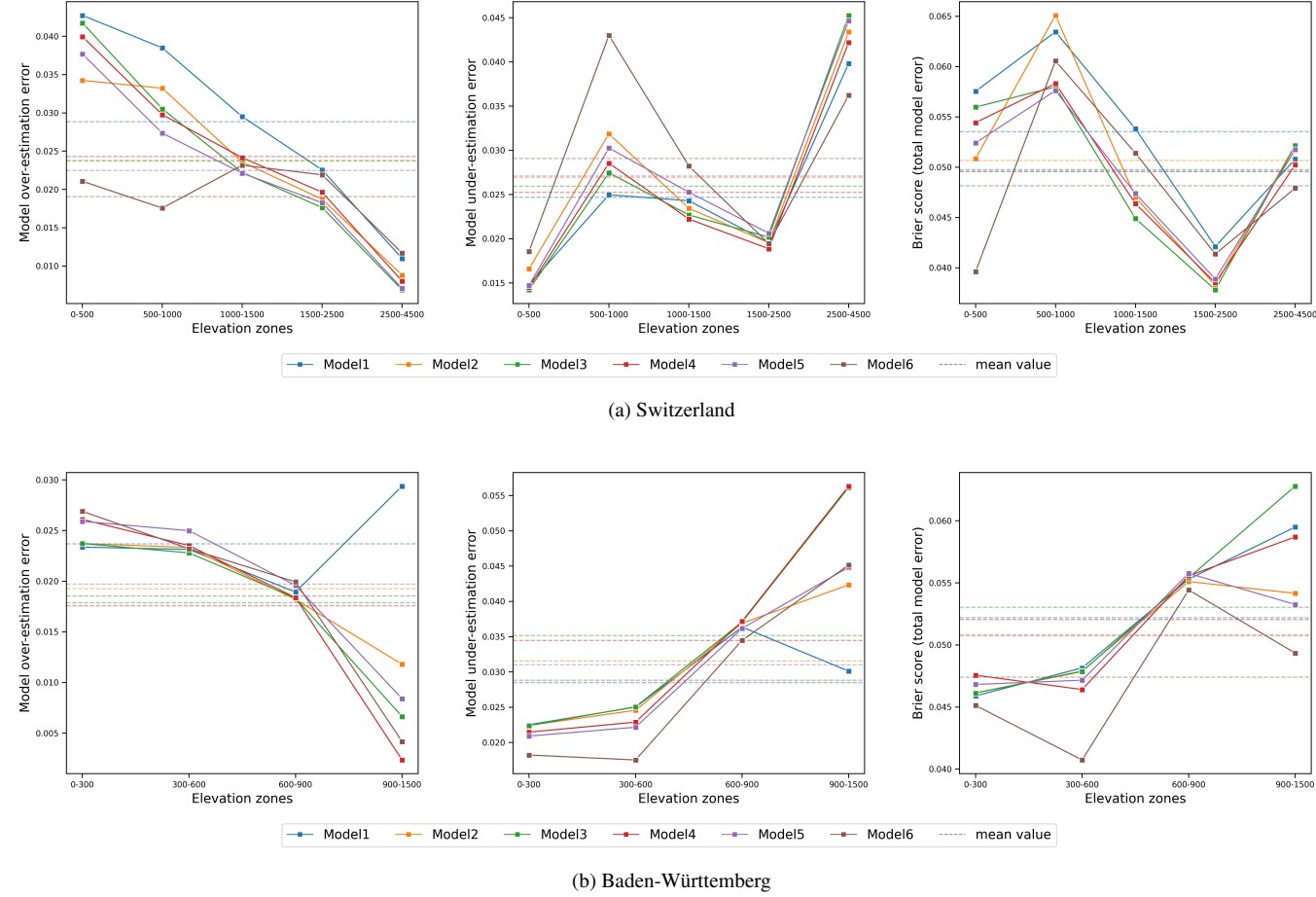

**Figure 5.** Model performances in different elevation zones (left: over-estimation error, middle: under-estimation error, right: total error (Brier-score), dashed: mean error for all elevation zones); (a) Switzerland (b) Baden-Württemberg

ages to be used as a reference series to calibrate against; and the snow detection threshold to define the simulated snow water
equivalent (SWE) as a snow pixel in the simulated snow cover distribution. Different NDSI thresholds ranging from 1 to 95,
cloud thresholds ranging from 10% to 90%, and detection thresholds from 0 to 5mm were considered for the analysis. A NDSI
threshold of '1' means that the MODIS images with pixel value of greater than 1 was considered as a reference snow pixel.
Likewise, a cloud threshold of 10% indicates that for a given calibration period, images with less than 10% cloudy pixels
would be selected for calibration. Similarly, a SWE threshold of 0.5mm would mean that the pixels with simulated SWE above
0.5mm were considered as a simulated snow-pixel. The detailed results are presented in the Appendix figures A1, A2 and fig.
6.

From the figures, it can be clearly inferred that a NDSI threshold of 30, cloud threshold of 10% and a SWE threshold above
0.5 mm could achieve the best performance in terms of Brier scores in Switzerland. Likewise, the results for BW show that, a

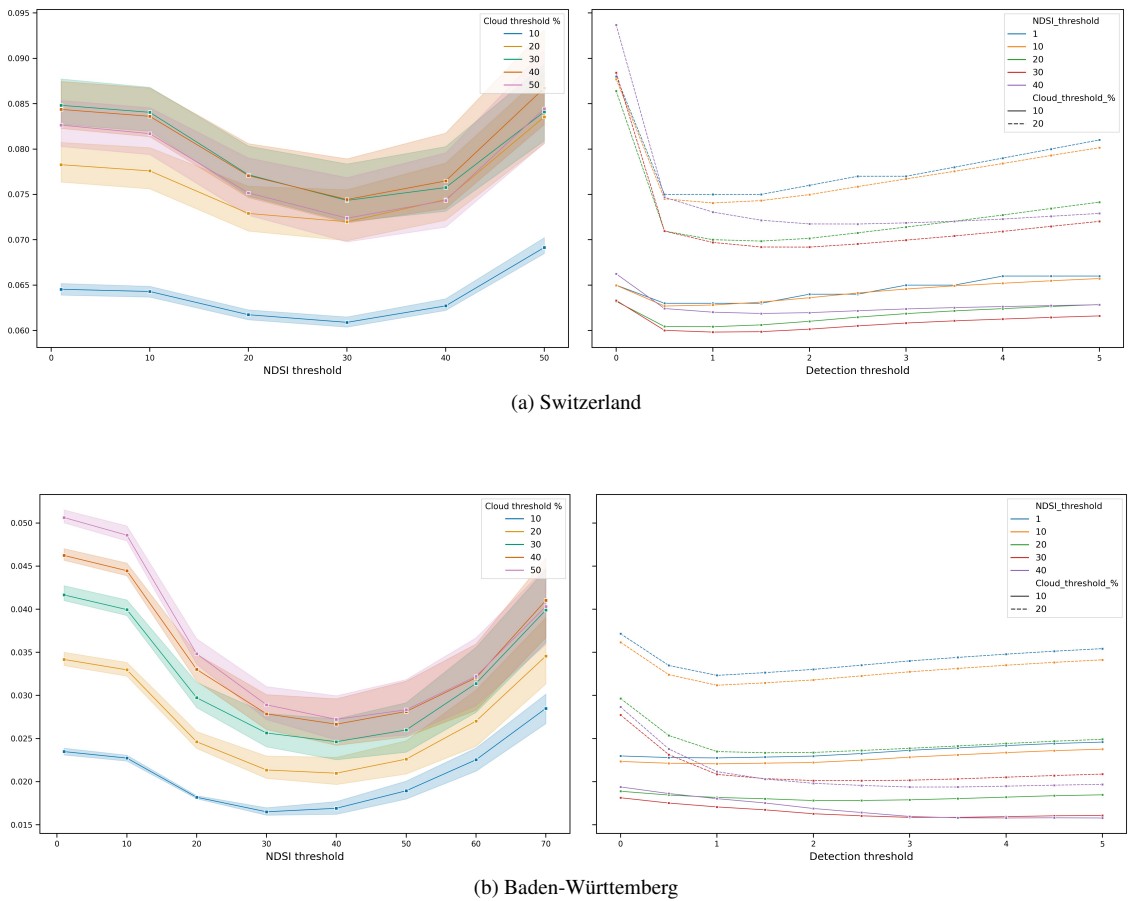

**Figure 6.** Model performance for different thresholds in (a) Switzerland (b) Baden-Württemberg; left plot: The shaded regions indicate the performance for different SWE thresholds. right plot: Dashed and solid lines respectively show performance for 10% and 20% cloud thresholds. Detection threshold in the X-axis level refers to the SWE threholds.

SWE detection threshold of 2.0 mm or above, coupled with the NDSI threshold of 30 and a cloud threshold of 10% gives the best simulation of the snow cover distribution. The results for both the regions are similar to each other which highlights the applicability of the adopted methodology in different snow regimes.

## 4.3 Sensitivity analysis on the selection of calibration periods

A sensitivity analysis on the selection of different calibration periods was also carried out for the snow season of 2012-2013. For this, the reference model was calibrated against sets of daily MODIS images representing different stages in a snow season, viz. whole season (Sep.-June for Switzerland and Oct. – May for BW), onset season (Sep. – Feb. for Switzerland and Oct. -Feb. for BW), melt season (Mar. – June for Switzerland and Mar. – May for BW), peak snow month (February for both) and the day with highest amount of cloud-free snow (Feb. 18 for Switzerland and Feb. 12 for BW). The thresholds were adopted as per

the earlier section i.e (NDSI: 30, Cloud %: 10, SWE threshold: 2.5mm for BW and NDSI: 30, Cloud %: 10, SWE threshold: 0.5 mm for Switzerland). The reference model was calibrated for all the aforementioned seasons using the ROPE algorithm.

1000 best parameter vectors were identified for each season. These parameters for each calibrated periods were then iteratively checked whether they were contained within the convex hull defined by the validation parameters, to identify the percentage of these contained parameters sets. Subsequently, the calibrated parameters were validated for each season to analyze the temporal transferability of the parameter vectors and identify the best calibration set that would work well with all the seasons/periods.

Figure 7 presents the validation of calibrated parameters in different seasons. This validation is expressed as the percentage

of the reference parameter vectors for each season/period contained by the hull defined by the validation parameter sets for Baden-Württemberg and Switzerland respectively. The figures indicate that the calibrated parameters for the whole season work well for different validation periods in both the regions, with a relatively higher containment of the calibration parameters in reference to the validation sets. In Switzerland, the melt season calibration exhibits a better temporal transferability as indicated by the higher proportions of the parameter vectors in each season whereas in Baden-Württemberg, the onset season calibration

shows higher transferability of the parameters.

Based on these results, it was concluded that the melt season calibration in Switzerland and the onset season calibration in Baden-Württemberg, imparted more robust parameter sets for both the study areas. The whole season calibration showed good agreement with other periods in both regions. Figure 8 provides a better understanding of the Brier score dispersion for the regions for these selected calibration periods.

In Baden-Württemberg, the onset season calibration is well validated for the peak snow season and the single day event. The melt season calibration is also comparable with the onset season performance. The whole season calibration has better performance for onset and melt seasons, but with a more uncertain validation for the peak snow month and single day image, as shown by the box-plot bounds. In Switzerland, melt season calibration shows a better validation for almost all the seasons. This indicates that the images available towards the end of the season can simulate the whole snow season very well. However,

the seasonal calibration performance remains similar as in the case of Baden-Württemberg.

## 4.4 Snow-melt model results at catchment level

The reference snow-melt model (Model 6) was also calibrated for two catchments in Baden-Württemberg and three in Switzerland on snow-cover distribution for the winter season throughout 2010-2015 (BW) and 2010-2018 (Switzerland). ROPE was used to calibrate the models and obtain 1000 sets of best performing parameter vectors for each catchment, based on the over-

all Brier-scores. The simulated snow-cover distribution was compared with the ones estimated by the HBV's snow-routine to assess the representation of snow accumulation and melt processes within. The 1000 HBV snow parameter sets were subsetted from the best performing parameter vectors obtained during ROPE calibration of the HBV model on discharge for each catchment. The comparison results are shown in fig.9. It is evident from the violin plots that the snow-melt models clearly out-perform the HBV snow routine in all catchments while estimating the snow-cover distribution estimation.The median

Brier-score values for the snow-melt models in all catchments are lesser than their counterparts. Moreover, the 1000 best Brier-score values for the snow-melt models depict a very narrow spread in contrast to a much wider spread from the HBV's snow

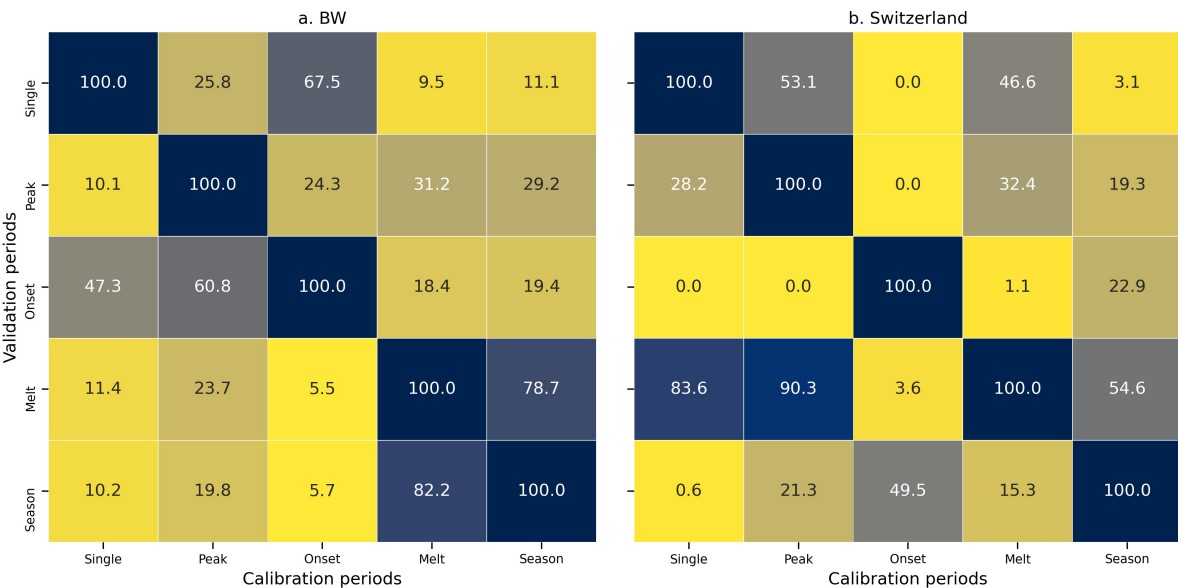

**Figure 7.** Percentage of calibrated parameter sets contained by the convex hull defined by the validation parameter sets

routine. This shows the uncertainty of the HBV during the simulation of snow accumulation and melt, which hints towards a compensating effect with other non-snow parameters. This spread is more pronounced in the Horb and Rottweil catchments, which are characterized by shorter duration snow. The results suggest that HBV's snow routine, when calibrated on discharge together with the other model parameters, is not able to capture the snow dynamics in BW region as compared to the spread in Swiss catchments with longer duration snow. This approach thus adds value to these regions as the calibrated snow-cover distribution provide a strong basis for estimating available water coming from snow, as these regions are dependent on the melt waters. The results, thus strongly indicate that the use of a standalone snow-melt model, in this study, provides a very stable and reliable representation of the snow-cover distribution and in turn the melt.

The boxplots showing the dispersion of the parameters are shown in Fig.10. The y axis shows the normalized parameter values based on their min-max range set for optimization. The boxplots indicate that apart from the radiation melt factor, $r_{ind}$, $T_{Mmax}$, $T_{Mmin}$ and to some extent $T_S$, all the parameter ranges are relatively stable and less sensitive to the model performance. These four parameters are however constrained by the objective function and it is understandably so because these parameters are more sensitive as they govern the appearance and disappearance of the snow.

## 4.5 Validation in hydrological models

The best performing parameter vector from the snow-melt models for each catchment was used to simulate the melt waters exiting the snow-regime, which was in turn, used in the modified HBV model to simulate the hydrologic implications of the melt as a standalone input. The modified HBV was calibrated on discharge for the whole timeseries for each of the catchments.

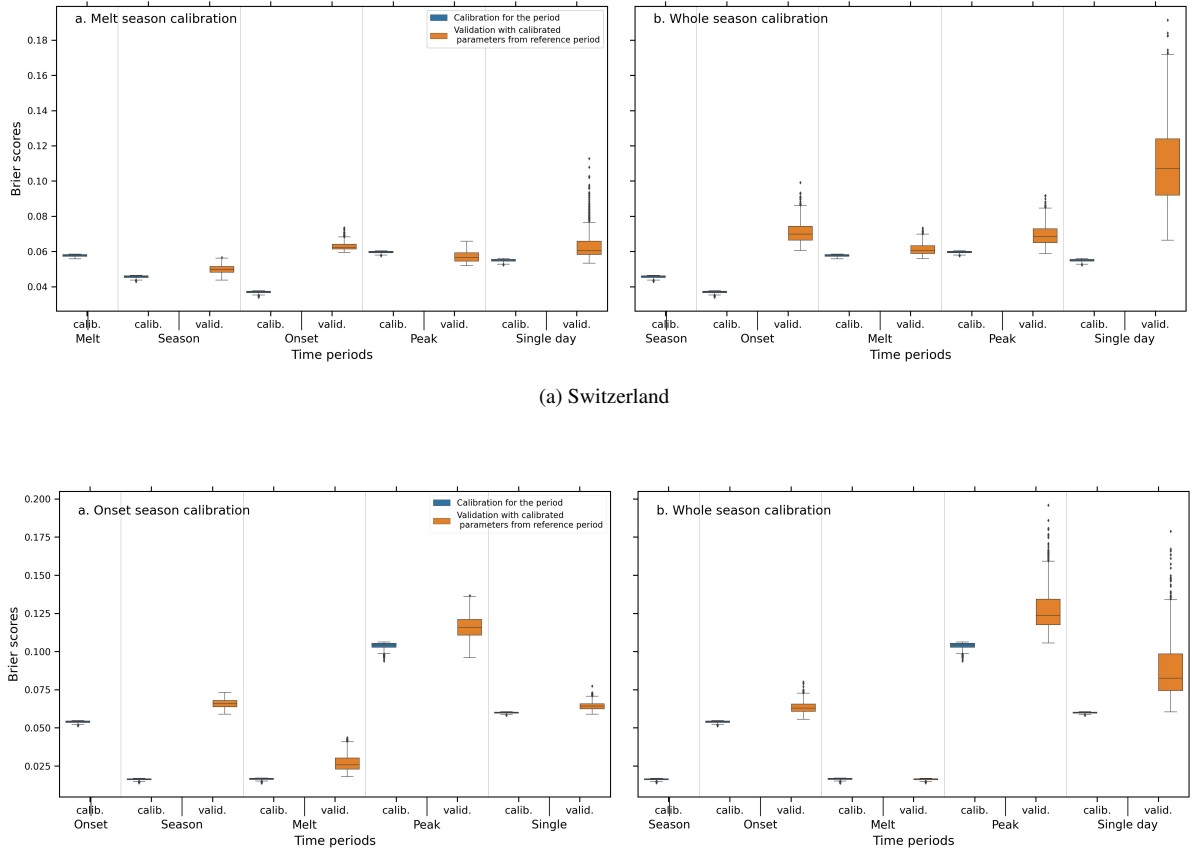

**Figure 8.** Model performance in the validation periods (a) Switzerland: (left plot: validation results for melt season calibrated parameters, right plot: validation results for whole season calibrated parameters) (b) Baden-Württemberg(left plot: validation results for onset season calibrated parameters, right plot: validation results for whole season calibrated parameters); X-axis labels 'calib.': calibration performance for the same season, 'valid.': validation performance using the reference season parameters

This was done with 3 iterations of ROPE. The standard HBV was also calibrated on the same discharge data, but with five ROPE iterations. With the ROPE calibration, 1000 robust parameter sets were identified for both hydrological models. The best NSEs were subsequently compared to assess the performance of the snow-melt models against the HBV. The results are shown in fig.11 and Table 4.

The results show that the addition of melt improves the hydrological model performance in each of the catchments, notably the most in the snow dominated ones in Reuss and Aare. The median NSEs show improvement in all of the catchments in the study domain. The NSE spread is also smaller in the in the case of modified HBV. This highlights uncertainty reduction, as the hull containing the equifinal parameter vectors becomes smaller as a result of lesser parameters to calibrate for the modified

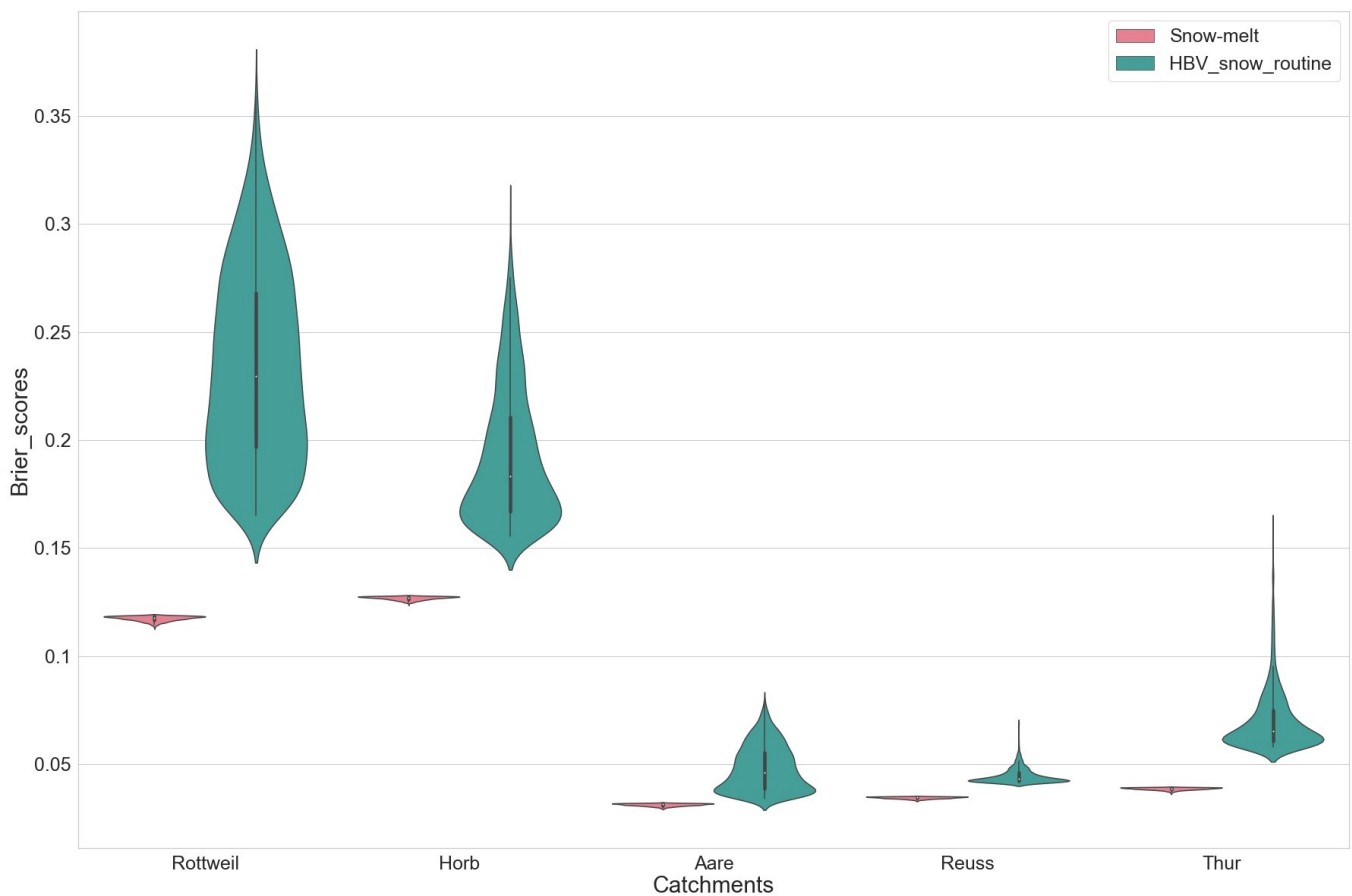

**Figure 9.** Violin plots comparing the performance of the reference snow-melt model in different catchments and HBV's snow routine in terms of Brier-scores; The height of the violin plots shows the range of Brier-scores whereas the shape of the curve depicts their density.

HBV variant. The results suggest that the improvement in model performance comes with a better computational efficiency and a better 'mimicry' of the snow accumulation and melt.

## 5 Discussion

It is a big challenge and a highly imperative one, to improve the snow-melt routines in widely and successfully tested rainfall-runoff models like HBV (Bergström, 2006; Girons Lopez et al., 2020). In this milieu, to evaluate the snow-processes based on spatial distribution of snow, we implemented an image-based pattern calibration approach using MODIS-inferred snow-cover distribution for a single day or a duration of snow season of a given year. The calibration was done based on pixel-based binary information ('1':'Snow', '0':'No snow') from the MODIS images in Switzerland and Baden-Württemberg. The

MODIS data available freely across the world and at a daily resolution, provides a plausible alternative to ground based data

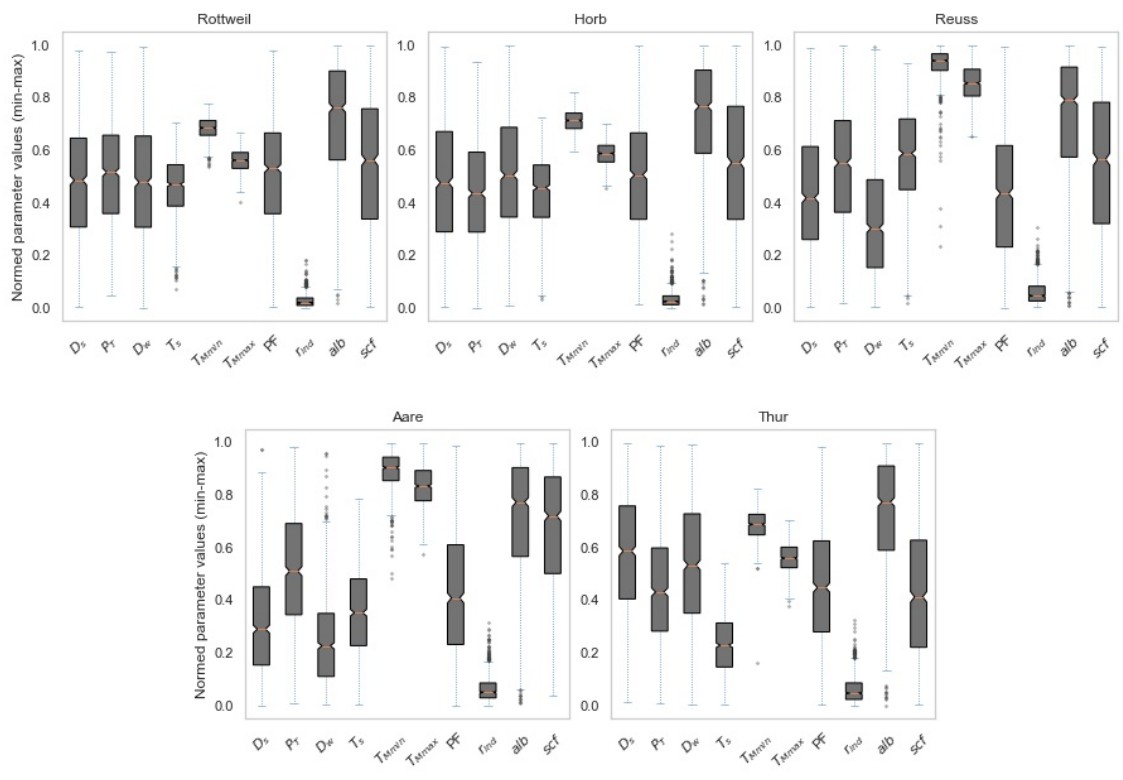

**Figure 10.** Reference Snow-melt model parameter dispersion for different catchments

**Table 4.** Comparison of HBV and Modified HBV NSE performance

|  | HBV NSEs | | | Modified HBV NSEs | | |
|---|---|---|---|---|---|---|
|  | Min | Max | Median | Min | Max | Median |
| Rottweil | 0.595 | 0.672 | 0.609 | 0.663 | 0.724 | 0.676 |
| Horb | 0.653 | 0.700 | 0.663 | 0.738 | 0.821 | 0.755 |
| Aare | 0.395 | 0.566 | 0.424 | 0.658 | 0.678 | 0.663 |
| Reuss | 0.635 | 0.779 | 0.656 | 0.781 | 0.796 | 0.785 |
| Thur | 0.702 | 0.768 | 0.712 | 0.731 | 0.776 | 0.739 |

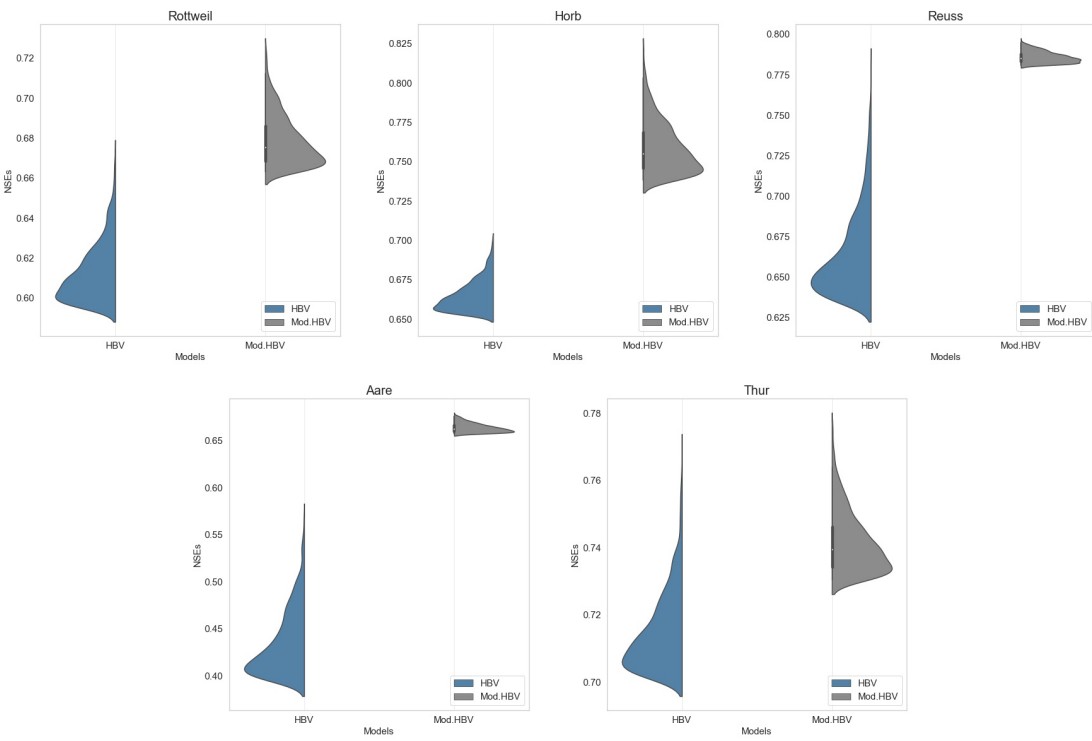

**Figure 11.** Performance comparison of the standard and the modified HBV models in terms of NSE in different catchments. The shape of the violin plots indicate the density plots for the 1000 best NSEs for each model.

for immediate verification of snow-melt models. Widely used and computationally simplistic temperature index models with low data requirement were considered in the study and were modified wherever possible, to gain enhanced model performance.

We found that all the model variants calibrated on a set of individual snow-cover images for the whole season were able to track the MODIS snow-distribution with very good accuracy. These models were evaluated based on a Brier-score which represents the total false recognition of snow and no snow pixels. Results from all the models are comparable and very close in simulating the snow-cover distribution. The models' performance in terms of over-, under- and total estimation errors were further scrutinized in different elevation zones. We observed that for Switzerland, characterized by the longer duration snow, all the models were found to show decreasing over-estimation error with increasing elevation, except Model 6 where the model error slightly increases in the mid-elevation zones. However, the model performances in terms of under-estimation errors tend to increase with higher elevations. We found that the mean overall error remained the lowest for Model 6 with the radiation component for Switzerland. In Baden-Württemberg, the overall error is the lowest for all elevation zones with the radiation-

based model. While some of the model variants were able to track the snow-cover distribution with better accuracy, we did not find a model-specific clear trend of prediction errors in different elevation zones. Based on the mean performance, Model 6 was deemed accurate enough to simulate the snow-cover distribution with a higher accuracy in identifying snow and no-snow pixels, reflected on the low Brier-score values. As a reference model for further analysis, we selected this relatively complex variant with daily potential clear-sky radiation, for both regions. The inclusion of radiation induced melt has been found to provide a more realistic spatial distribution of melt rates (Hock, 1999). However, it is to be noted since this approach is flexible enough to work with different snow-melt modules which can simulate the snow-cover distribution, we selected the relatively complex radiation-based model to assess the added value of the robust binary data selected for calibration purpose.

We also carried out sensitivity analysis on the different thresholds used in the study to differentiate between the snow and no-snow pixels. We found out that a NDSI threshold of 30, a cloud percentage threshold of 10% and a SWE threshold of above 2.0mm perform with very good accuracy for a seasonal calibration where all MODIS images fitting the cloud threshold criteria are used to calibrate the models in both regions. A closer look into the analysis suggests that, for Baden-Württemberg, NDSI thresholds of 30 and 40 give similar results when the SWE threshold of higher than 2.0mm is used with 10% cloud thresholds. In Switzerland, NDSI threshold of 30 showed the best performance with SWE thresholds higher than 0.5mm. However, it is to be mentioned that, this analysis was done assuming MODIS as the observed snow-cover distribution. Earlier studies on the NDSI thresholds (Tong et al., 2020; Härer et al., 2018) have suggested that a threshold of 40 can be used for robust estimates of snow cover from MODIS. However, in reality, the spatial detail of MODIS and the associated uncertainty (Tong et al., 2020) makes it harder to know where the snow actually is. Coupled with this, is the uncertainty associated with the precipitation data. These added uncertainties make it harder to recommend a definite threshold for future applications. The adopted calibration technique was found to be more sensitive towards the NDSI threshold, particularly in Switzerland, as it can be observed from Appendix fig. A1. In contrast, Appending fig. A2 shows that the sensitivity to NDSI is not that significant in Baden-Württemberg. It is thus, our observation that a NDSI threshold of 20-50 can be considered to demarcate the snow/no-snow information for MODIS based snow-melt simulations, along with a SWE threshold of 0.5 - 5mm for longer duration snow conditions and 2-5mm for relatively shorter duration snow conditions, without much loss in model performance. Nester et al. (2012) have also identified a SWE thresholds of 2.5mm as optimum for error analysis with MODIS SCA. A cloud detection threshold of 10% was found to be the ideal threshold for the selection of MODIS images for calibration although our methodology is not that sensitive in terms of cloud coverage as can be seen in Annex figures. This highlights the fact that this calibration on binary pixel information can also be done on patches of snow-covered pixels free from clouds, which makes this approach very flexible. However, different studies have found cloud threshold to be a critical factor during the evaluation of model errors. Şorman et al. (2009) adopted a cloud threshold of 20% for a multi-variable hydrological model calibration. Nester et al. (2012) suggested a cloud threshold of 80% as there were no significant differences in model errors for <50% and 50-80% cloud cover in their study.

The sensitivity analysis on the selection of dates of the MODIS images for calibration showed that the selection of the dates has a very strong impact on the model performance and in turn on transferability of parameters. The melt season calibration in Switzerland and the onset season calibration in Baden-Württemberg were found to have the best validation in other seasons

and periods. With a more clearly defined melt season in Switzerland, we found out that the images towards the end of the snow season were adequate enough to simulate the snow processes within the snow season with good accuracy and these calibrated parameters were found to be robust as shown by the spread of Brier-scores in the validation periods (refer fig. 8). However, in Baden-Württemberg, which is characterized by a distinct onset but a more uncertain melt season (lesser snow availability), the onset season calibration was observed to be more robust and transferable to other validation periods.

On a catchment level, three catchments in Switzerland and two in Baden-Württemberg were selected. The snow-melt model was calibrated using multiple images for winter seasons throughout the time period to obtain a set of 1000 robust parameter vectors. The temperatures demarcating snow and melt onset and the radiation melt factor were deemed sensitive to the overall Brier-scores. The Brier-scores from these 1000 simulations were compared with the ones estimated with parameters in the HBV's snow routine. The calibrated snow model significantly outperforms the HBV's snow model calibrated on discharge in all of the catchments. This has also been discussed by Udnæs et al. (2007) and Parajka and Blöschl (2008) in their respective studies that calibration on SCA in addition to runoffs improved the snow model efficiency. Further, our results clearly show that the uncertainty in simulation of the snow-cover distribution is significantly reduced when using a dedicated snow-melt model, as the snow parameters in the HBV calibrated on discharge are compensated with other non-snow parameters. This compensation leads to a more uncertain representation of the snow accumulation and melt dynamics. This decrease in uncertainty with a simplified calibration approach using freely available data is noteworthy as the calibrated parameter vectors are more robust as compared to the ones calibrated on discharge. Similar findings have been pointed out by Riboust et al. (2019) where they discussed the increased robustness of the snow model parameters when SCA was added into the calibration. Since this methodology implements the calibration at pixel level on binary ('snow', 'no-snow') information instead of depths, relatively complex snow-melt models are also calibrated with more robustness and without over-calibration of the models. This can be attributed to the robust data selected for calibration and the spatial extent of the satellite images used for simulation.

It is to be noted that this study doesn't intend to assess the performance of any specific hydrological model predictions, however, aims to assess the performance of the melt outputs of snow-melt models calibrated on snow-cover distribution in a basic hydrological model. We deem it important to point out that though two of the Swiss catchments, Aare and Reuss are glaciated, the glacial-melt was not taken into account. This was mainly because of the MODIS' limitation to identify the glaciers (Muhammad and Thapa, 2020) and to avoid further parameterization of the hydrological model to include glacier component. Since, our aim is to evaluate the snow-melt model calibration, both hydrological models were calibrated on equal terms with similar approach with results comparable to each other. We found good improvement in the NSE performance in all the catchments while using the modified HBV. Parajka and Blöschl (2008) also concluded that the median runoff model efficiency increased when MODIS data was used for the calibration in comparison to a traditional discharge calibration. Bennett et al. (2019) concluded in a similar tune with their results showing MODIS fSCA improving the internal snow timings as well as the hydrological simulations.

Our results further show that the NSE dispersion is also reduced with the modified HBV simulations indicating a reduction in model uncertainty as both the sets of parameters required for calibration as well as the set of equifinal parameters becomes smaller. This gain in performance comes with better computational efficiency, as the modified HBV calibration converges

faster. This allows for incorporation of additional parameters in the independent snow-melt models to better represent the snow dynamics, as opposed to the fact that additional parameters in hydrological models impose further complexity during calibration. With a dedicated snow-melt model, this can be achieved as it is relatively quicker to calibrate on the images and the resulting melt can be used in the hydrological models for efficient calibration. Di Marco et al. (2021) also concluded that a combination of MODIS fractional snow-cover area and streamflow data led to a reduction of predictive uncertainty of a hydrological model thereby leading to sharper and reliable flow simulations. Furthermore, the strength of this approach lies in the simplicity, spatial flexibility and global availability of the model input data which can be very useful for snow-melt and hydrological predictions in data scarce regions.

## 6 Conclusions

The study presented a methodology to evaluate a daily MODIS snow-cover based calibration in identifying distribution patterns of snow in snow-dominated regimes in Switzerland and Baden-Württemberg region in Germany. Specifically, different model modifications were employed to assess the improvement in the simulation of snow-distribution with lesser input requirements. By calibrating at pixel level on binary information, relatively complex snow-melt modules can be calibrated with more robustness as the uncertainty associated with calibration data is reduced as it is often with the case of snow-depth or snow-water equivalent based calibration. From our results we can conclude the following:

(a) It was observed that the methodology does well in mimicking the snow-cover distribution in the regions with relative higher accuracy with all the models.

(b) For the study regions, NDSI thresholds and SWE thresholds for snow/no snow differentiation respectively for the MODIS and simulated snow distribution were identified along with the best cloud percentage threshold option critical for the selection of MODIS images for calibration.

(c) Depending upon the snow regimes, the results suggest that a set of MODIS images within a period during the snow season are adequate to adeptly simulate the snow-cover distribution for the whole season.

(d) Comparison of the snow-melt model's performance with the HBV's snow routine shows that the uncertainty in the representation of snow-accumulation and melt processes can be reduced with a standalone calibration of a snow-melt model as HBV calibration on discharge usually exhibits compensating behavior with other non-snow parameters. The improvement in model performance can be deemed for 'a right reason' with a better representation of the underlying snow processes.

(e) The calibration using readily available images used in this method offers adequate flexibility, albeit the simplicity, to calibrate snow distribution in mountainous areas across a wide geographical extent with reasonably accurate precipitation and temperature data, especially in data scarce regions, with parameters estimated with MODIS. The other data used for the snow models can be derived from publicly available digital elevation models.

The reduction in model uncertainties, primarily with the snow-distribution estimation and with the discharge simulation, adds value to provide improved conceptualization of the temperature-index model routines and further potential model updating in future works. Furthermore, this approach is not dependent on the choice of hydrological models as it can be extended to any hydrological model that can identify the snow-cover distribution.

*Data availability.* The precipitation and temperature data were obtained from the Climate Data Center of the German Weather Service (DWD; https://opendata.dwd.de/climate_environment/CDC, last access: 15 February 2021) (DWD, 2021) and the Swiss Federal Office of Meteorology and Clima-tology (MeteoSwiss; https://gate.meteoswiss.ch/idaweb, last access: 21 Decemeber, 2020)(MeteoSwiss, 2020). The MODIS snow-cover images were downloaded using the Earth Data Search tool (https://search.earthdata.nasa.gov, last access: 19 Feb 2021). The DEM was obtained from http://srtm.csi.cgiar.org.

*Author contributions.* This study is a part of DG's doctoral research supervised by AB. The study was conceptualized by AB and DG, and was implemented by DG. Both authors contributed to the writing, reviewing and editing of the paper.

*Competing interests.* The authors declare that they have no conflict of interest.

*Acknowledgements.* The authors would like to acknowledge Deutscher Akademischer Austauschdienst (DAAD) for the doctoral research scholarship which encompasses this study.

*Financial support.* This open-access publication was funded by the University of Stuttgart.

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

# Appendix A: Sensitivity analysis for different thresholds

## A1: Switzerland

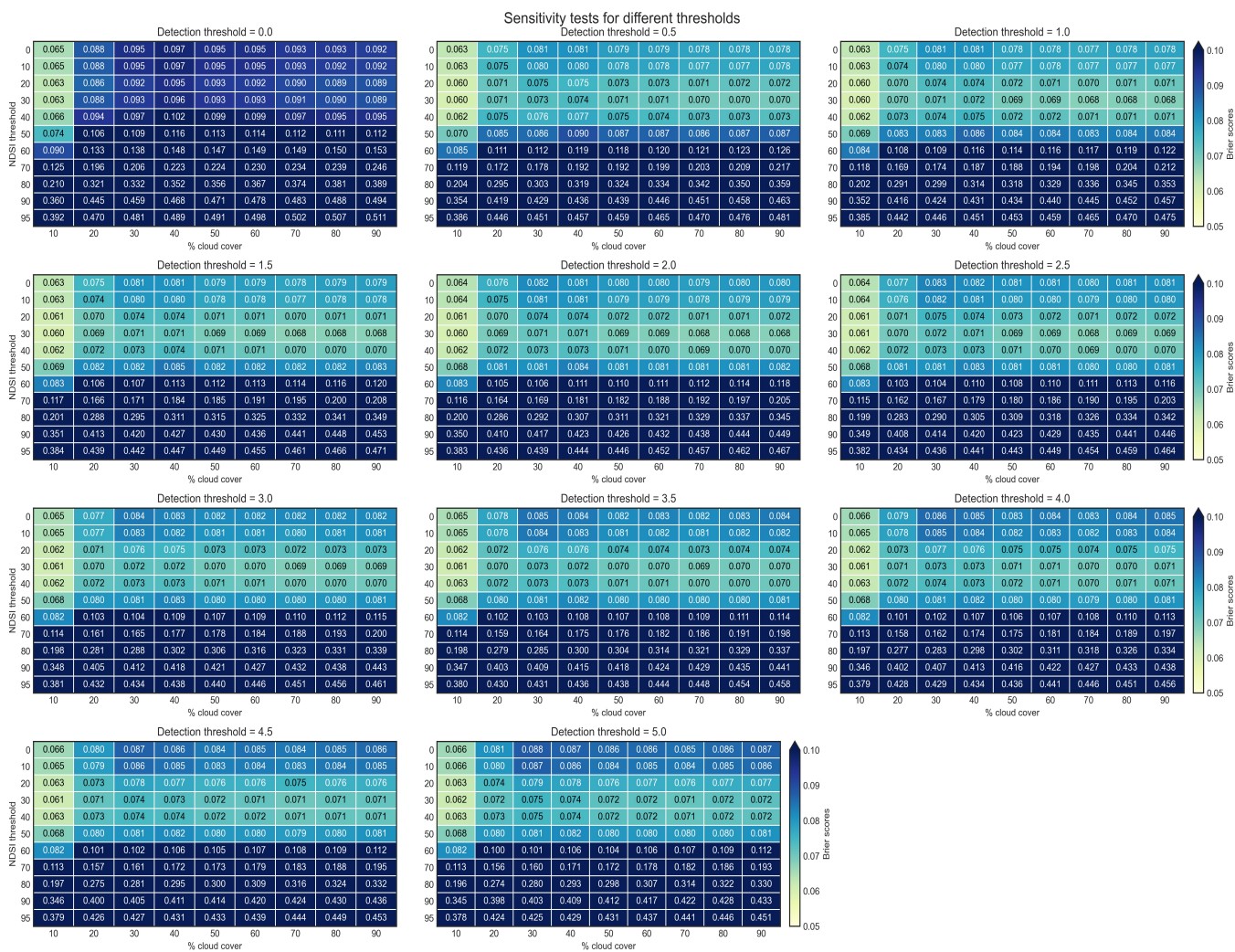

**Figure A1.** Sensitivity analysis results for NDSI, Cloud percentage and SWE thresholds in Switzerland; Detection threshold in the figure refers to the SWE thresholds.

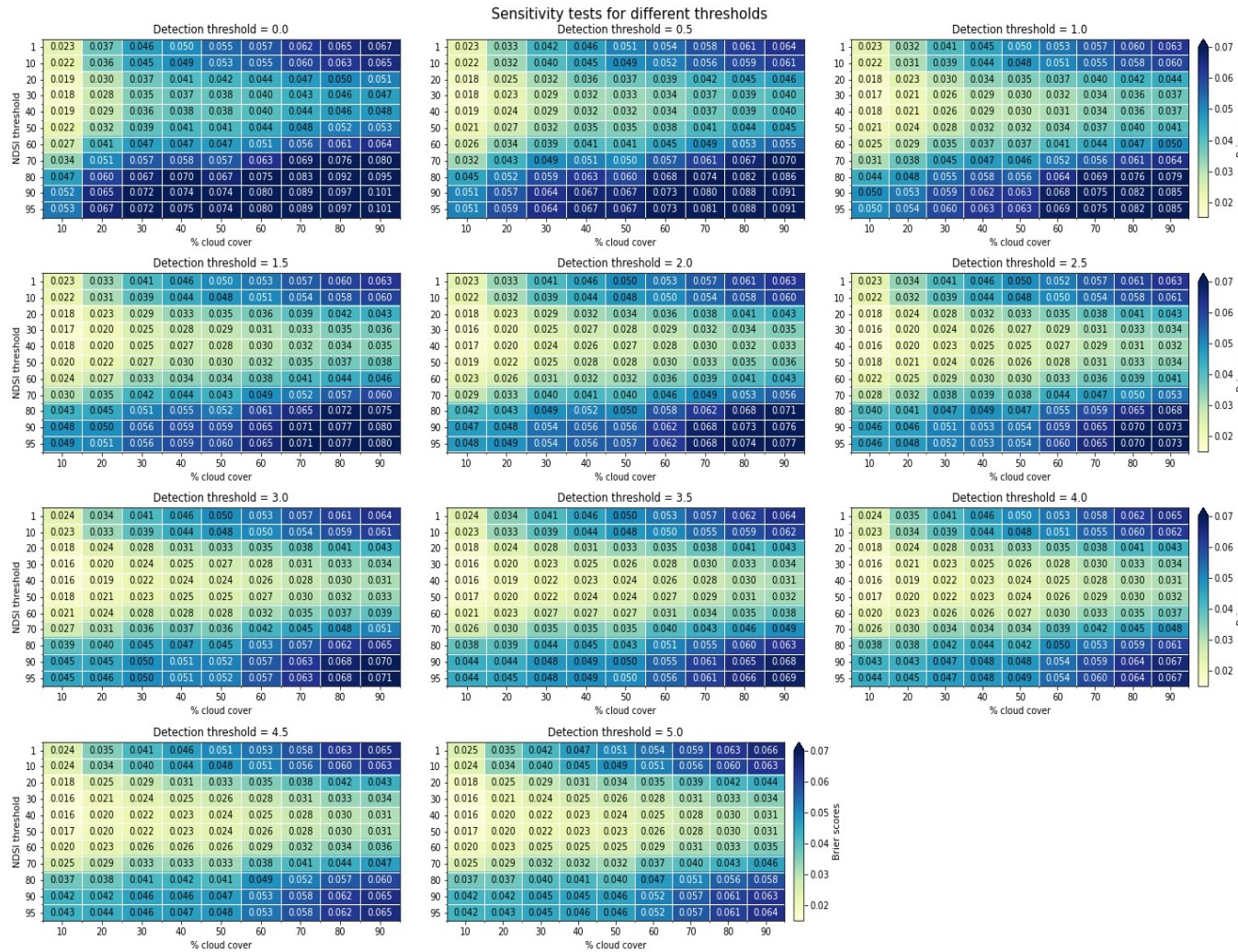

**Figure A2.** Sensitivity analysis results for NDSI, Cloud percentage and SWE thresholds in Baden-Württemberg; Detection threshold in the figure refers to the SWE thresholds.