# Peer review of "Development and parameter estimation of snow-melt models using spatial snow-cover observations from MODIS"

_Hydrology and Earth System Sciences, 2021_

## Author Comment (AC2)

**Response to Reviewer #2**

We would like to extend our thanks to Bettina Schaefli for taking her time to thoroughly read the paper and give her critical remarks.

While there are many valid points in her review, there are quite a few points where we do not agree.

*I read the manuscript and the review by reviewer 1. I completely agree with his point 1 that the literature review is incomplete. For each investigated modification, we would need a reference for who has done this before. Besides a review of how others tried to improve the degree-day method, we also need reference to pattern calibration in hydrology, i.e. for calibration on other patterns (e.g. evaporation).*

This is statement related to fundamental problems of present publications. The second author of the paper serves himself as an editor and has the opinion that the introduction of the papers increased significantly in the past 20 years. The introductions are becoming medium quality mini review papers, which are unimportant for people who are working on the topic, and of little use for those who are unfamiliar with the topic. They serve mainly for the increase of citations.

The purpose of scientific publications is to share new ideas and experiences with fellow scientists. Review papers are an exception, but they should be written by experts with sufficient experience, and not by PhD students. The review process should focus on the novelty and applicability presented in the paper. Due to the present evaluation of scientists, it is unfortunately not likely that this tendency to fill a paper with a large number of references mainly in the introduction will change, as promotions depend on the number of papers and citations. We will however, follow the suggestions of the reviewers to rewrite an introduction with extensive references, but we wanted to express our discomfort.

*I furthermore believe that something went wrong with the subsection numbers. It would be good to use classical sections (intro, case study, method, results, discussion. conclusion) and to avoid mini-subsections.*

This can be done, we will reformulate the sections and avoid the sub-subsections.

*Overall, the idea to calibrate the simple snow routine (composed of an accumulation routine and a degree-day melt routine) on single-day snow image is indeed interesting. The results show that a calibration on a single snow image gives as good results as calibrating the snow routine directly on streamflow (the reference HBV). And furthermore, calibrated parameter values for different years are relatively stable.*

These are our main findings.

*While it can in principle be interesting to test different snow accumulation and melt approaches, I am not convinced that the presented approach is*

*able to provide any valuable insights. The reason is that the provided model performance numbers do not tell us whether one model is really better than another one; what does the increase of NSE by 0.002 actually mean? Even an increase from 0.82 to 0.85, is this significant?*

In this case the calibration of the snow model was not done with the help of the hydrological model, but instead independently on satellite images. If one would calibrate the snow model using the hydrological model the uncertainty would increase due to the fact, that other parameters of the model may compensate for problems in the snow module. It is always advantageous to try to calibrate submodels independently and on other quantities. This way the model parameters are likely to become good because of the right reasons. This will also contribute to the reduction of the equifinality of the model parameters.

Further snowmelt models are not only important for hydrological modelling. For example the estimation of snow amounts are very important for water management. The suggested model could be inverted to provide estimates of snow amounts in remote regions.

An increase of the NS value of the hydrological model was not the goal, it is thought to show that the independently achieved snow module is good enough to become an non-discharge calibrated part of the hydrological model. The difference in the NS could be significant. It is not the difference which decides on the significance of the change of an NS value. As an analogous example consider the correlation coefficient: In a normal case with a sample size of 365, the increase of a correlation from 0.85 to 0.89 or from 0.94 to 0.96 is highly significant while a seemingly much higher increase of the correlation of 0.25 to 0.35 is not significant (all on 95 % significance level).

$$z = \frac{\ln \frac{1+\rho_1}{1-\rho_1} - \ln \frac{1+\rho_2}{1-\rho_2}}{2\sqrt{\frac{1}{n_1-3} + \frac{1}{n_2-3}}}$$

For the NS values the idea is similar, but unfortunately the calculation of significance of the change of the NS value would require independence of the errors which is not the case. Further, there is no good model which could describe the dependence of the model errors. Treating this topic goes far beyond the scope of this paper. But for sure, one should not decide on the significance whether two NS values are different or should not be done only on the basis of difference of the NS values.

Fisher, R. A. (1921). On the Probable Error of a Coefficient of Correlation Deduced from a Small Sample. Metron, 1, 3-32.

*The advantage of calibrating the snow parameters directly on snow images as opposed to calibratinjg on discharge (which is still required to calibrate the other model parameters) is not clearly discussed.*

A discussion on this will be added to the updated version of the paper. As

mentioned above, the advantage of independent calibration of parts of the model contributes to the reduction of uncertainty and to the identification of modeling problems.

*There are only three case studies, on reflecting intermittent snow, the other one reflecting the typical build up of a seasonal snow cover. Is calibration on snow of a particular interest in one case or the other?*

Calibrating on snow cover is interesting in both cases. For intermittent snow cover it has direct influence on discharge, while for build up and melting it can contribute to the estimation of snow amounts.

*Finally, I believe something went also wrong with the choice of the Swiss catchment. I do not know which outlet you have chosen, but it seems to be (given the high discharge) after the outflow of a large lake in Luzern, which means that you emulate the behavior of the lake through the hydrological model. Any effect of the snow cover / melt will be strongly smoothed out by the lake volume.*

Thank you for the hint - we will use another catchment.

*For all above reasons, I cannot recommend a revision of this paper for publication in HESS. The material could be turned into an interesting contribution but would require substantial work, which I believe goes beyond major revisions.*

This is a very disappointing statement as we believe that we presented a new and useful method, which could be useful for others, and your critics are mainly concerning the presentation, and very little additional computations are needed. Thus the revision of the paper could be done in a reasonable time.

**Detailed comments:**

*- the water balance equation of the snow cover is missing; how does liquid precip enter the equation? ie. is liquid water immediately added to the snowpack outflow?*

The snow-cover is only used for calibrating the snow models. Considering a grid cell, if the cell has snow water equivalent (SWE) greater than 0.5mm, it is reclassified as '1' and '0' if not. This would be our modeled binary snow cover distribution. The model is then calibrated using the MODIS snow cover distribution independently (also binary '1' for cells with snow and '0' for cells without) as the reference. The liquid water is not considered in the snow models. However, this is immediately added to the melt water outflow in the truncated HBV model.

*- melt does not increase because of liquid water falling on the snowpack, this is a physical misconception; in wet conditions (rainfall or high relative humidity), the heat transfer from the atmospher to the snowpack is higher than in dry conditions; it is this heat transfer that increases the melt and not the advected heat via rainfall entering the snowmpack*

*(which is very small); and of course the liquid water content of the snow-pack increases with incoming rain but we do not know how the liquid water content of the snow cover is modelled here (point above)*

Thank you for the suggestion. We agree that the melt induced by the latent heat carried by the rain entering the snowpack, albeit less, contributes to melt. This section will be slightly rephrased. Please refer above for the last query.

*- list of paramters / variables are unusal, I would put in text format*

The feedback is well-noted.

*- we need information on the used catchments (classical catchment caracteristics, including location, size, etc).*

This will be done in the revision.

*- remove unnecessary digits in the tables*

The digits will be rounded in the tables in the revised manuscript.

*- how to you define winter? what are snow days? with snowfall or snow on the ground?*

In this study, we have assumed winter as a mean snow-season for the regions. The winter season was considered from October to April in Baden-Wuerttemberg and September to June for Switzerland. 'Snow days' are defined as the days with available snow (above 0.5 mm SWE) in the regions till it disappears. This will be defined clearly in the revision.

*-Any conceptual water-streamflow transformation model has to be calibrated with the water input. If you change the input, you have to re-calibrate the other model parameters. Here, the input with the different snow models is probably only marginally different, but this should be mentioned anyway; and if the input is only marginally different, how can we conclude that the snow models lead to different performance.*

As already mentioned above, the calibration of the snow model was not done with the help of the hydrological model, but instead independently on satellite images. However, we will re-calibrate the truncated HBV in the revision with the melt from snow-models as the standalone inputs. The results will then be discussed in the revised manuscript.

*- avoid multi-letter variable or parameter names*

The feedback is well noted. The letters were added to provide a clear definition of which variable is being used in the models to avoid confusion, as the models take different inputs.

*- NSE values for precipitation against elevation are not really interesting; NSE depends on the underlying signal seasonality, which varies with elevation and with region*

The cross validation of temperature and precipitation is not a novel finding. It was only included to partly quantify the possible error of the input data. We intend to shorten this part of the paper.

*- reference for Residual Kriging?*

This will be added in the revised document.

*- time frame of simulations, which time period is used for the precip and temperature stations*

For Baden-Wuerttemberg, the time period used was from 2010 to 2015 and for Switzerland it was from 2010 to 2018. This will be added in the revised document.

*- did you account for gauge undercatch for precip measurements during winter (snow)? if not, this will most likely strongly underestimate snowfall*

The gauge undercatch for precipitation measurements during winter was not accounted for. We could try including a correction factor in the revision.

*- snow season in Switzerland can start in September but only at very high elevations, where melt season continues in July*

Thank you for this suggestion. We have focused on the period from September to June for Switzerland to demarcate the 'snow' season.

*- I would not discuss model updating and (real-time) forecasting; this is a very different topic and would require more references and in-depth discussion;*

Well noted. This will be revised.

---

## Author Response (AR1)

**General Response**

We would like to thank both reviewers for taking their time to thoroughly review our paper and providing crucial feedback. We have streamlined our manuscript in the new revised version with detailed review of similar approaches carried out in the past. As per the reviewers' suggestions, we have tried to formulate a better narrative to the study. The review, we believe, has helped to improve the manuscript.

Three more catchments were added to analyse the efficacy of our approach. We have further discussed about how independent calibration helps in improving the snow-model performance thereby reducing the uncertainty related to snow melt simulation as compared to the hydrological model simulations, and how it can add value to discharge simulations.

**Response to Reviewer #1**

We thank Juraj Parajka for taking his time to carefully read the paper and providing critical remarks.

**General comments**

*This study presents an approach for calibrating different degree-day snowmelt approaches by using MODIS snow cover data. The second aim is to examine different degree-day variants for snowmelt simulations and calibrate or validate them using satellite snow cover data. The approach is tested in two regions (Baden-Württemberg in Germany and Switzerland). The results indicate a slight increase in overall NSE runoff performance and a better NSE performance during the winter period.*

*I read with interest the manuscript because we did numerous similar experiments in the past (and recently). I have to say that the manuscript presents some interesting and novel experiments, but as a whole, it is not ready for publication in its current form. The main reasons for such assessment are:*

*The Introduction section needs to be improved. In its current form, it is not specifically presenting which approaches are already available, what the research gaps are and how this research goes beyond existing studies? There are numerous studies (for example, please see some references below, and references cited in these studies) investigating and comparing different degree-day snowmelt models and studies investigating calibration of conceptual hydrologic models (their snow part) to MODIS snow cover data. The introduction needs to clearly present the research done so far and to formulate what the novel scientific contribution of this study is. In my opinion, a comparison of existing degree-day models is not novel. Nor a general use of MODIS snow cover data in hydrological modelling. Still, I think the study presents some interesting approaches which can be turned into novel research objectives, such as how many and which MODIS images are needed for robust calibration of conceptual snowmelt*

*models.*

The introduction of the paper was certainly not complete, but this is not a review paper. In our understanding, the purpose of a non-review paper is to provide new ideas and not to give a complete picture of the state of the art. If the paper contains facts that are already known and not referenced, then this should be pointed out by the reviewers. Recently publications have extremely long review like introductions and number of cited papers increased significantly in the past years. In the opinion of the second author, this did not improve the quality of the publications.

However, with constructive feedback from the reviewer, have reformulated the Introduction section including a literature review of previous studies and the novelty of our approach in form of research objectives.

*The structure of the document/story is not easy to understand, and the clarity of the presentation can be improved. If the study's main aim is to propose some novel approach/method, then I would suggest presenting it first and describing the study region and data later. This will allow the reader to understand the novelty and eventually to apply the general approach to other regions/models. I would also suggest presenting a general strategy at the beginning clearly. This will create a storyline and improves the clarity of the presentation. In its current form, there are many subsections and the order reads more like a summary of all technical works done but does not present clearly what the novel scientific contribution/research hypothesis is.*

Thank you for this suggestion. We have tried to reconstruct the narrative in a more focused way. The general strategy, as suggested, has been modified. The subsections are removed wherever possible and the structure of the paper is streamlined to better exhibit the research idea and the results.

*The study needs to be more focused on the novel contribution. I'm not sure how the interpolation and its cross-validation contributes to the novel scientific findings in the field of using satellite data form model calibration? Perhaps the crossvalidation can be presented only in a supplement. The more interesting point is to analyse which MODIS images are needed for robust model calibration. I do not understand why not use all available images, particularly for model validation? How sensitive are presented results to the selection of dates of MODIS images? There should be a more detailed analysis and evaluation for supporting the results and interpretations made. It is also not clear why not to use the concept of the HBV for simulation of snowmelt accumulation and melt. Why is it needed to separate the degree-day part and then link it back with the hydrological modules instead of using it together (i.e. to calibrate only the snow module first and then apply the complete model)?*

The cross validation of temperature and precipitation is not a novel finding. It was only included to partly quantify the possible error of the

input data. We have shortened this part of the paper and omitted from the Results section. Furthermore, we have added catchment based snowmelt models using the winter images for the entire time-period used in the study. In our opinion, the separation of the calibration and validation of the snow model from the hydrological model makes sense because of many reasons. Some of them are: (i) different hydrological models may use the same snow model (ii) an independently calibrated snow model reduces the uncertainty of the model parameters as no compensation of the model errors is possible through model parameters (reduced equifinality) (iii) snow models may be used individually for the estimation of available resources.

*The discussion of the results is not comprehensive. It will be interesting to link the findings with similar studies calibrating the hydrologic models by using MODIS or comparing different variants of degree-day models.*

Thank you for this suggestion. The discussion and concluding sections are extended with literature review in the revised manuscript.

*I believe the manuscript presents an interesting topic and can be an interesting contribution, but it needs a very substantial revision and extension.*

We would like to thank you for your encouraging words. We have done our best to improve the paper in its revised form.

**Specific comments**

*Which MODIS version is applied?*

Version V6 for both MODIS Terra (MOD10A1.006) and MODIS Aqua (MYD10A1.006) were used in the study. This was added in the revision in Line 115.

*Kriging. Was the spatial correlation model (semi-variogram) fitted separately for each day?*

Yes, the semi-variogram models were fitted for each day.

*Radiation based model: how was the Linke coefficient estimated. Does it vary seasonally?*

The Linke coefficient was set at a constant value of 3.0 (close to the annual mean for rural-city areas) for the model. Seasonal variation was not accounted for in the study but it could be an interesting addition. Instead, a diffusion factor (0.2 for clear sky to 0.8 for overcast conditions) was used to account for the diffusion.

*Cross-validation of interpolation. Leave-one –out crossvalidation is typically used to camper different interpolation methods. Were the residuals smaller than obtained by some other interpolation method? How do the resulting maps compare with existing gridded (precipitation, air temperature) products provided by MeteoSwiss or DWD?*

The goal of presenting the cross validation results was to give some information on input parameter uncertainty. The presented results suggest the Residual Kriging worked better than other methods for precipitation and External drift Kriging worked better for temperature data. However, the detail explanation of kriging and its results, due to the lesser relevance to the overall study, has been removed in the manuscript.

We have not compared the Kriged surfaces with the DWD or Meteoswiss gridded products yet.

**Response to Reviewer #2**

We would like to extend our thanks to Bettina Schaefli for taking her time to thoroughly read the paper and give her critical remarks.

While there are many valid points in her review, there are quite a few points where we do not agree.

*I read the manuscript and the review by reviewer 1. I completely agree with his point 1 that the literature review is incomplete. For each investigated modification, we would need a reference for who has done this before. Besides a review of how others tried to improve the degree-day method, we also need reference to pattern calibration in hydrology, i.e. for calibration on other patterns (e.g. evaporation).*

This is statement related to fundamental problems of present publications. The second author of the paper serves himself as an editor and has the opinion that the introduction of the papers increased significantly in the past 20 years. The introductions are becoming medium quality mini review papers, which are unimportant for people who are working on the topic, and of little use for those who are unfamiliar with the topic. They serve mainly for the increase of citations.

The purpose of scientific publications is to share new ideas and experiences with fellow scientists. Review papers are an exception, but they should be written by experts with sufficient experience, and not by PhD students. The review process should focus on the novelty and applicability presented in the paper. Due to the present evaluation of scientists, it is unfortunately not likely that this tendency to fill a paper with a large number of references mainly in the introduction will change, as promotions depend on the number of papers and citations.

We have however, followed the suggestions of the reviewer and extensively rewritten an introduction with section with references pertaining to the use of Remote Sensing in modeling and prior approaches to improve the models in its revised form. We have also added references to each model formulation as suggested in the Methodology section.

*I furthermore believe that something went wrong with the subsection numbers. It would be good to use classical sections (intro, case study, method, results, discussion. conclusion) and to avoid mini-subsections.*

The sections are reformulated and mini-sections removed in the revised manuscript.

*Overall, the idea to calibrate the simple snow routine (composed of an accumulation routine and a degree-day melt routine) on single-day snow image is indeed interesting. The results show that a calibration on a single snow image gives as good results as calibrating the snow routine directly on streamflow (the reference HBV). And furthermore, calibrated parameter values for different years are relatively stable.*

Thank you for your feedback. The findings are presented in a better way in the revised document.

*While it can in principle be interesting to test different snow accumulation and melt approaches, I am not convinced that the presented approach is able to provide any valuable insights. The reason is that the provided model performance numbers do not tell us whether one model is really better than another one; what does the increase of NSE by 0.002 actually mean? Even an increase from 0.82 to 0.85, is this significant?*

In this case the calibration of the snow model was not done with the help of the hydrological model, but instead independently on satellite images. If one would calibrate the snow model using the hydrological model the uncertainty would increase due to the fact, that other parameters of the model may compensate for problems in the snow module. This has been well-documented in the revised manuscript for better understanding. Our results support the hypothesis that it is always advantageous to try to calibrate submodels independently and on other quantities. This way the model parameters are likely to become good because of the right reasons. This will also contribute to the reduction of the equifinality of the model parameters.

Further snowmelt models are not only important for hydrological modelling. For example the estimation of snow amounts are very important for water management. The suggested model could be inverted to provide estimates of snow amounts in remote regions, as an extension to this research.

An increase of the NS value of the hydrological model was not the goal, it is thought to show that the independently achieved snow module is good enough to become an non-discharge calibrated part of the hydrological model. The difference in the NS could be significant. It is not the difference which decides on the significance of the change of an NS value. As an analogous example consider the correlation coefficient: In a normal case with a sample size of 365, the increase of a correlation from 0.85 to 0.89 or from 0.94 to 0.96 is highly significant while a seemingly much

higher increase of the correlation of 0.25 to 0.35 is not significant (all on 95 % significance level).

$$z = \frac{\ln \frac{1+\rho_1}{1-\rho_1} - \ln \frac{1+\rho_2}{1-\rho_2}}{2\sqrt{\frac{1}{n_1-3} + \frac{1}{n_2-3}}}$$

For the NS values the idea is similar, but unfortunately the calculation of significance of the change of the NS value would require independence of the errors which is not the case. Further, there is no good model which could describe the dependence of the model errors. Treating this topic goes far beyond the scope of this paper. But for sure, one should not decide on the significance whether two NS values are different or should not be done only on the basis of difference of the NS values.

Fisher, R. A. (1921). On the Probable Error of a Coefficient of Correlation Deduced from a Small Sample. Metron, 1, 3-32.

*The advantage of calibrating the snow parameters directly on snow images as opposed to calibrating on discharge (which is still required to calibrate the other model parameters) is not clearly discussed.*

A discussion on this has been added to the revised version of the paper. As mentioned above, the advantage of independent calibration of parts of the model contributes to the reduction of uncertainty and to the identification of modeling problems. However, we have re-calibrated the modified HBV with melt outputs from the snow-model and explained in detail in the results and discussion.

*There are only three case studies, on reflecting intermittent snow, the other one reflecting the typical build up of a seasonal snow cover. Is calibration on snow of a particular interest in one case or the other?*

Calibrating on snow cover is interesting in both cases. For intermittent snow cover it has direct influence on discharge, while for build up and melting it can contribute to the estimation of snow amounts. We have added 3 more catchments to the study to validate our findings at catchment levels.

*Finally, I believe something went also wrong with the choice of the Swiss catchment. I do not know which outlet you have chosen, but it seems to be (given the high discharge) after the outflow of a large lake in Luzern, which means that you emulate the behavior of the lake through the hydrological model. Any effect of the snow cover / melt will be strongly smoothed out by the lake volume.*

Thank your this critical comment. We have used the upstream catchment of Reuss at Seedorf instead, in the revised version. We apologise for the mistake. Furthermore, results from two more Swiss catchments, Aare at Brienzwiler and Thur at Andelfingen were also added to the revised paper.

*For all above reasons, I cannot recommend a revision of this paper for publication in HESS. The material could be turned into an interesting contribution but would require substantial work, which I believe goes beyond major revisions.*

This is a very disappointing statement as we believe that we presented a new and useful method, which could be useful for others, and your critics are mainly concerning the presentation, and very little additional computations are needed. We have done the revision in a reasonable amount of time.

**Detailed comments:**

*- the water balance equation of the snow cover is missing; how does liquid precip enter the equation? ie. is liquid water immediately added to the snowpack outflow?*

The snow-cover is only used for calibrating the snow models. Considering a grid cell, if the cell has snow water equivalent (SWE) greater than 0.5mm, it is reclassified as '1' and '0' if not. This would be our modeled binary snow cover distribution. The model is then calibrated using the MODIS snow cover distribution independently (also binary '1' for cells with snow and '0' for cells without) as the reference. The liquid water is not considered in the snow models. However, this is immediately added to the melt water outflow in the modified HBV model.

*- melt does not increase because of liquid water falling on the snowpack, this is a physical misconception; in wet conditions (rainfall or high relative humidity), the heat transfer from the atmospher to the snowpack is higher than in dry conditions; it is this heat transfer that increases the melt and not the advected heat via rainfall entering the snowmpack (which is very small); and of course the liquid water content of the snowpack increases with incoming rain but we do not know how the liquid water content of the snow cover is modelled here (point above)*

Thank you for the suggestion. We agree that the melt induced by the latent heat carried by the rain entering the snowpack, albeit less, contributes to melt. This section has been rephrased with the approach done by Bàrdossy et al. 2020. Please refer above for the last query.

Bárdossy, A.; Anwar, F.; Seidel, J. Hydrological Modelling in Data Sparse Environment: Inverse Modelling of a Historical Flood Event. Water 2020, 12, 3242. https://doi.org/10.3390/w12113242

*- list of paramters / variables are unusal, I would put in text format*

The feedback is well-noted. However, for the clarity of the presentation, as many parameters are involved, we have for now kept it as it is.

*- we need information on the used catchments (classical catchment caracteristics, including location, size, etc).*

This has been added in the Study area and data section.

*- remove unnecessary digits in the tables*

The digits were rounded in the tables in the revised manuscript.

*- how to you define winter? what are snow days? with snowfall or snow on the ground?*

In this study, we have assumed winter as a mean snow-season for the regions. The winter season was considered from October to April in Baden-Wuerttemberg and October to June for Switzerland. 'Snow days' are defined as the days with available snow (above 0.5 mm SWE) in the regions till it disappears. The winter periods are defined clearly in the revision in Lines 312-313

*-Any conceptual water-streamflow transformation model has to be calibrated with the water input. If you change the input, you have to re-calibrate the other model parameters. Here, the input with the different snow models is probably only marginally different, but this should be mentioned anyway; and if the input is only marginally different, how can we conclude that the snow models lead to different performance.*

As already mentioned above, the calibration of the snow model was not done with the help of the hydrological model, but instead independently on satellite images. However, we have re-calibrated the modified HBV in the revision with the melt from snow-models as the standalone inputs. The results clearly discuss the implications and reduction in uncertainty in the revised manuscript.

*- avoid multi-letter variable or parameter names*

The feedback is well noted. The letters were added to provide a clear definition of which variable is being used in the models to avoid confusion, as the models take different inputs.

*- NSE values for precipitation against elevation are not really interesting; NSE depends on the underlying signal seasonality, which varies with elevation and with region*

The cross validation of temperature and precipitation is not a novel finding. It was only included to partly quantify the possible error of the input data. We have shortened this part of the paper.

*- reference for Residual Kriging?*

This has been added in the revised document in Line 149.

*- time frame of simulations, which time period is used for the precip and temperature stations*

For Baden-Wuerttemberg, the time period used was from 2010 to 2015 and for Switzerland it was from 2010 to 2018. This was clearly mentioned in the revised document in Lines 312-313.

*- did you account for gauge undercatch for precip measurements during winter (snow)? if not, this will most likely strongly underestimate snowfall*

The gauge undercatch for precipitation measurements during winter was not accounted for in the first submission. However, we have added a correction factor to account for the combined gauge undercatch and vegetation interception for all the snow-melt models and the HBV. Please refer to Eq. 2a.

*- snow season in Switzerland can start in September but only at very high elevations, where melt season continues in July*

Thank you for this suggestion. We have focused on an average period from October to June for Switzerland to demarcate the 'snow' season on a more regional level.

*- I would not discuss model updating and (real-time) forecasting; this is a very different topic and would require more references and in-depth discussion;*

Well noted. This has been omitted from the revision.

---

## Author Response (AR2)

**General Response**

We would like extend our gratitude to both reviewers for their valuable feedback and suggestions, which have allowed us to scrutinize our work further and refine it accordingly. We have agreed to most of their feedback and incorporated in the revised manuscript wherever possible. The review, we believe, has helped to improve the manuscript.

Specifically, sensitivity analyses on different thresholds and calibration dates in the study regions have been added to provide new insights regarding our approach in the study regions. Below are our responses to the comments/feedback.

**Response to Juraj Parajka**

We thank Juraj Parajka for his continued feedback and constructive critics on our work, which was highly imperative to streamline our manuscript as well as the presentation. Below are our responses to his comments.

1. - *The motivation and context of the study can still be improved. In its current form, the story is not focused and clear. For example, the Introduction states, "Various modeling and measurement techniques are currently in practice ..." and "Prior studies on the comparison of snow models have highlighted the higher reliability of physically based approaches". It is not clear how these statements support the need to examine simple degree-day approaches in the study? Or, it is not clear how the referenced studies about multiple objective calibrations connect to the study objectives. How are calibrations using scatterometer or evapotranspiration data related to the context of this study? The Introduction can still be more focused and improved.*

   The mentioned statements were put into the introduction section as a problem statement to bolster our methodology which uses readily available data throughout the world in contrast to the data intensive and site specific complex approaches. We agree that multiple objective calibration part was longer than required, but they were put there for an overall picture of how MODIS is being used in hydrological predictions. The latter part was a suggestion in the first round of review. However, the introduction part has been streamlined and made more focused in the revised manuscript.

2. - *Formulation of the novel scientific contribution and research objectives need to be more precise. Is the main objective development of a flexible snow-melt module? I do not think so. Or at least, the results, in their current form, do not refer to a new flexible snowmelt routine.*

   The feedback was well-noted and the novelty and objectives sections are clarified in the Introduction section.

3. - *The formulation of the main novel contribution is also somewhat general and vague. In general, the stepwise calibration of hydrologic models (i.e. calibrating the snow module*

*first, the runoff generation in the next step) is not new. Also, using MODIS for step-wise calibration was already examined and evaluated in previous studies (see, e.g. Szeles et al., 2020, DOI:10.1029/2019WR026153). In my opinion, the novelty is mainly in proposing/testing/examining/evaluation standalone spatial patterns of snow cover (from MODIS) for calibration of the conceptual snow module. The formulation and presentation of the generality of findings can be improved. The study applies numerous assumptions and thresholds and it is not clear how these affect the results and how to set up the thresholds/assumptions in future studies. For example, the study assumes NSDI threshold equals 1 for snow cover classification. The previous versions of MODIS were based on a globally fixed threshold equalled 40. The recent study of Tong et al (2020, https://doi.org/10.1016/j.jhydrol.2020.125548) examined this threshold and found that 40 works well for Austria unless some more detailed information is available. But the threshold=1 likely does not provide the best mapping accuracy compared to observed snow depth. How was this threshold determined? What is the advice for readers in using this threshold in future applications? Other thresholds are, for example, used for snow cover mapping evaluation. Why 2.5mm and 0.5mm? Why 60% of valid pixels? How sensitive are the results to these settings? How should the readers setup these thresholds in future studies? The most important but unclear point is, in my opinion, the definition of which day to use for model calibration? Why selected dates? Again how sensitive are the results to this selection? What to advise to the readers in future applications?*

Following this suggestion, we did sensitivity analyses on different thresholds and sensitivity of dates. The results are available and discussed in the revised manuscript.

4. *- The need for cloud removal techniques is not clear. If only one image is used for model calibration, is this method/step needed? Why this order of steps?*

The cloud removal techniques were employed to obtain as much data as possible for calibration, using simpler yet effective steps. The model analysis have been changed to sets of images within a season, so the cloud removal technique comes into play in the revised analysis. However, for a single image, the selection can be done so that the image with least amount of clouds can be picked.

5. *- The description of the calibration procedure does not provide sufficient detail to reproduce the experiment. Particularly the ROPE approach and uncertainty analysis can be described in a more detailed way.*

The ROPE steps were added into the revision. Please refer to Lines 298 - 310.

6. *- The comparison of different degree-day approaches is very brief. In my opinion, it is a missed opportunity to describe and evaluate the differences between the approaches in more detail. It is not clear how the results affect the date selection for model calibration. But are the pixels with the better model performance of model 6 linked to accounting for the radiation input? I will be very interested to see a much more extensive comparison of the approaches. It will be very interesting to see why and where are some approaches more robust than the others. As it is already commented above, the generality of findings based on only one image*

*is rather low and a more detailed assessment will be very interesting to see. For model evaluation, it will be very interesting to see also whether and which models overestimate or underestimate the MODSI snow cover and where?*

We would like to thank you for this suggestion. The model performances in terms of over-, under- and total estimation errors were scrutinized under different elevation zones for both BW and Switzerland and added in the revised manuscript.

7. - *The section about the results of transferability of model parameters does not read well. The method is not introduced in the methods section and it is also not clear how this is linked with the main objectives of the study. Perhaps this part can be a separate study as the forecasting is a quite specific topic and the manuscript in its current form does not provide a clear context for this analysis. I think that more detailed comparisons of the degree-day models and sensitivity analysis of how different thresholds and calibration date selection affect the results will provide already a nice and compact analysis worth to be published.*

The mentioned section was omitted. This was to show the spatio-temporal flexibility of the calibration which can be presented in a separate study.

8. - *Finally, the Discussion can provide some advice on setting up the analysis (definition of the thresholds, selection of dates) in different regions or time periods. Such lessons will improve the presentation of the generality of findings.*

Thank you for the suggestion. The discussion and conclusions have been revised.

**Response to the Anonymous reviewer**

We would also like to thank the anonymous reviewer for taking the time to carefully read our paper and providing critical remarks and suggestions. We have tried to address all the queries and incorporate his/her valuable suggestions. Below are our responses:

**General comments**

*- The main objective of the study was to develop relatively simple extended degree-day snow models driven by freely available snow-cover images. Authors see the novelty of their research in independent calibration of the snowmelt models on snow cover images which allows standalone estimation of associated parameters and thus a better representation of the snow processes. Output from these snow models were later used as input data in modified HBV model for streamflow simulation in five selected catchments in Germany and Switzerland.*

*First, it should be noted that the paper has been reviewed by two reviewers before and authors created a new version of the manuscript. After reading the reviewers comments and authors replies, it becomes clear that the study has been significantly revised. Nevertheless, I did not base my review on the earlier reviews, and rather tried to comment the revised study without bias.*

*In my opinion, authors did an interesting work. I certainly agree that the focus on testing different variants of degree-day models and their calibration against snow cover area using MODIS data is important, although not fully novel. Similarly, the de-coupling of snow routine from the selected hydrological model and its standalone calibration might bring some new insight on calibration procedures and model equifinality, although many hydrological models are nowadays calibrated using more variables next to streamflow (SWE, snow cover, groundwater levels etc.). Therefore, I found the study important and particularly novel. I thus agree with previous reviews that the study is worth publishing in HESS. However, I have several specific comments and questions regarding the methodology approach and quality of presentation. These comments should be carefully addressed before I can recommend the manuscript for publication. I only partly checked the original manuscript (before the revisions), so I hope I will not be in contradiction with initial reviews.*

- We thank you for your kind response and critical review.

**Specific comments**

1. *- In my opinion, introduction section still needs partial improvement. Especially part within lines 45-66 looks like a list of studies containing just a short description without synthesis of individual information and results. I read the comments of the reviewers in the first round of reviews and their concerns regarding the introduction as well as authors response. Therefore, I do appreciate that authors extended the introduction section, but in my opinion, it resulted only in a partial improvement. Although I understand previous authors argumentation about writing long reviews with citing unrelated studies (and with a deep respect to the second author experience), I still think that it should be possible to write a relatively short and focused introduction section which shows the state of the art of the topic and research gaps which helps the readers to understand what's going on in the topic. Therefore, I would like to encourage the authors to improve the introduction section once again and to better relate*

*individual information to each other.*

As per the review, the introduction section has been modified.

2. - *Two study catchments, Reuss and Aare have some percentage of area covered by glaciers, whereas the glaciation cover for Aare is relatively high (15.5%) and thus the glacier melt considerably influences catchment runoff. Was glacier routine somehow included in the HBV model structure which authors used to simulate streamflow? I did not find this information in the text and thus it seems that glacier routine was not used. If true, I am not sure to what degree the simulations reflect the real observed values (at least for the Aare catchment). Could this somehow influence results interpretation? While I think the missing glacier component is not a problem for snow models and related results interpretation, I think it might be important for interpretation of results related to "standard" HBV and "modified HBV" (although authors assessed NSE values just for cold season months, I assume the simulation itself were done over the whole period 2010-2018). The most straightforward solution would be to include the glacier component for the two glaciated catchments (at least for the Aare catchment), or at least I would like to ask the authors to carefully address this point in discussion section.*

The objective of this study was to assess the performance of MODIS based calibration on snow accumulation and melt processes. We did not evaluate the Glacial-melt due to the MODIS limitations in identifying the glaciers. Both hydrological models were calibrated without the glacier component for uniformity. Nevertheless, we consider your suggestion as a very pertinent feedback. We added some lines in the discussion section (Lines 570 - 574).

3. - *L 313-317: It is not fully clear to me how exactly authors proceeded when creating the variants of a hydrological (HBV) model. If I understood correctly, authors created six HBV model variants for each catchment (which were named as "modified HBV"). These six HBV variants did not contain snow routines since snowmelt simulations resulting from previously defined six snow routines were directly used as input data to the HBV model. Last variant (seventh) was just a "true" HBV with its snow routine in its original structure (which is partly different than other snow routines, due to, e.g., including water holding capacity and refreezing). Is it right? If yes, then please, consider reformulation of the respective method part to be clearer. The fact that you used HBV snow routine to compare it with other six snow routine variants became clear only from results section to me (mainly from Fig. 7). Therefore, to improve the clarity of methods section, I would suggest modifying it such as you will describe seven variants of the snowmelt model (Model 1 – Model 7); the six you already have and the last (seventh) representing the original HBV snow routine. The seventh snow routine variant should be calibrated in the same way as the other six variant and comparison will be plotted in Fig. 7. In my opinion, using this procedure would make it clearer how well/badly your snow routines perform compared to the original snow routine structure implemented in HBV.*

Thank you for this suggestion. We have revised these sections accordingly. We presented six snow-melt modules, out of which the radiation based model was selected for further analysis as it reported the lowest Brier-score. This model was used to simulate the melt

using the best parameter vector and this melt was included in a modified HBV (without snow routine) as a standalone input. The other hydrological model was the standard HBV model. These two hydrological models were then calibrated on discharge at catchment level. To differentiate the differences of calibrating on discharge only, we subsetted the HBV's snow routine parameters (1000 best) and used it in HBV's snow routine to simulate the snow-cover distribution and calculate the 1000 Brier-score values, which were compared with the radiation based model's Brier-scores as shown in the mentioned figure (Figure 9 in the revised manuscript). The point here was to show how calibrating on discharge can over-compensate during individual processes simulation. Nevertheless, we have added some clarity in the Model calibration and validation sections.

4. - *Related to comment above, can be the differences between "modified" and original HBV (shown in Fig. 9) attributed to separated calibration of the snow routines or rather to different model structures of the snow routines or both? Can you differentiate between these two influences? Maybe it would be methodologically clearer, if you calibrate the "modified" HBV model against discharge separately for all seven snowmelt inputs (as describe in my comment above) as a first step (this is what you probably did). This way you can better compare which snow routine performs better when implemented in the HBV model. In the second step, you may select just "modified" HBV model with snowmelt inputs from separately calibrated HBV snow routine (snow model variant 7 as suggested in my comment above) and compare it with calibration of the original HBV model. This way, the first step shows the differences between individual snow routine structures (including original HBV snow routine), the second step shows the advantages/disadvantages of separate snow routines calibration compared to "normal" calibration (just against discharge) of the complete HBV model.*

As mentioned in the discussion section and the response above, our goal was to implement a MODIS based calibration on snow-melt modules, and specifically not to assess the performances of the widely used hydrolgical models like HBV. We just wanted to reiterate our finding that calibrating a hydrological model solely on discharge can have compensating effect on individual sub-processes. We did not use the snow-routine from HBV for snow-distribution calibration. However, it might be good step for future implementation of our approach where we can categorically present the findings in the manner you suggested. For this paper, we have just evaluated the snow-distribution simulation based on a discharge-based and a distribution-based calibration.

5. - *Important question is also whether the model performance should be assessed using NSE only. Current best practise is to use more criteria to make the results more robust. Would results interpretation change in case you will use different objective criteria (logarithmic NSE, volume error, etc.)? With this comment I come a little back to what was mentioned by Reviewer 2 in the first round of reviews, and it is to what degree the values of a single objective function (NSE in this case) could really tell us whether the one model is better than another (especially in case of small differences).*

This is a valid point. However, as we have pointed out earlier and in the first round of reviews, the objective here was to evaluate the snow-cover based calibration approach and its outputs.

Different objective functions can be added, but the aim was to see if it would add value to the underlying sub-processes. Maybe in the future studies, we can add more criteria including a multi-variable constrained calibration to evaluate the performance of the hydrological model. However, we wanted to show that the parameter set required for calibrating a hydrological model becomes smaller when using a step-wise approach of calibrating the snow module separately. This allows us to identify a more robust set of parameters along with the parameters related to the snow-processes estimated for a 'right reason' with a better representation of underlying snow processes, thereby gaining similar or better performance in terms of discharge simulation. This gain, albeit smaller or even similar, is a gain nonetheless arising from a better representation of the inherent snow accumulation and melt processes.

6. - *In my opinion, the discussion section should be improved since it seems to me that it is not clearly linked with results. It is certainly the matter of personal preferences, but I prefer using the results section just for results description and basic interpretation related to a single figure/table described, and everything which goes beyond a single figure interpretation (it means the results interpretation in a wider context of all your presented results and other literature) should be placed in discussion section. In this respect, the discussion section should be comparable to results in its extend and it may follow (not necessarily) similar structure as the structure of result section.*

The discussion section has been modified to better connect with the results.

7. - *Overall, the text is often difficult to follow since there are a lot of unclear statements, and it is not often clear how exactly authors proceeded (see also points above). This is also the case of some of figure and tables which are not clearly linked with the text, and they do not provide the reader with all needed information, such as informative caption or correct legend. Please see also my detailed comments in the list below. Maybe my comments and criticism stem just from these unclear issues rather than from real problems in methodological approach and results interpretation. Anyway, I would like to encourage the authors to go carefully through the text and try to make the text clearer and more consistent.*

Thank you for this comment. We have tried our best to present our findings with more clarity in the revised manuscript.

**Technical corrections**

1. - *L15: Please use "Nash-Sutcliffe efficiency" instead of NSE in abstract.*

This has been corrected.

2. - *L17: Two full stops at the end of the sentence.*

This has been corrected.

3. - *L 88-93: I would omit this paragraph since I found it too general. In fact, this is how all scientific papers are organized, thus, the specific description is not necessary here.*

We have omitted this paragraph in the revision.

4. *Fig. 1: Legend for elevation for the three inset figures (study area) seems not correct to me. As far as I can recognize, the colour scale is continuous in these small inset maps, thus the legend should be displayed accordingly (there aren't only four or five colours in figures, right?). Besides, in case of intervals are used for the colour scale (which is, to my knowledge, the best cartographic practise), the legend should be displayed without spaces between individual coloured rectangles. Additionally, use "Elevation [m a.s.l]" for the respective legend caption and add graphical scale (for all inset maps and the main map).*

The figure is updated in the revised manuscript.

5. *- L 99, 101: please use "m a.s.l." instead of "masl" (please check also other potential occurrences in the text whenever relevant.*

This has been done.

6. *- L 101: The highest point of Switzerland is 4634 m a.s.l. (Dufourspitze, Monte Rosa massif). This should be also reflected in legend of Fig. 1 (the last number in the legend). In this context, I would prefer the "real" highest point rather than the highest cell of the DTM raster you used to create the map.*

This has been modified.

7. *- Please use correct unit conventions (km2, m a.s.l.)*

This is corrected.

8. *- Section 2.2: Why not to use official Meteoswiss and DWD gridded products (which are available for much finer spatial resolution than your interpolations)? Was it because you needed also Tmax and Tmin while official gridded products were created only for daily Tmean and P? Or was there any other reason? Please clarify shortly.*

We wanted to align our interpolation to MODIS schema and we needed Tmax and Tmin as well, as interpolated grids. Furthermore, we wanted to test different interpolation techniques especially for precipitation. So for uniformity, we did not use the Swiss or German gridded products.

9. *- L 182: Authors mentioned that their "Basic Degree-day model" (Model 1) is the same model as implemented in HBV. However, this is not fully true since the snow routine implemented in the HBV model accounts also for liquid water holding capacity (which delays the water release from snowpack and thus directly influences daily SWE values) and refreezing (which has usually only a small effect on SWE calculation, at least at seasonal temporal scales). Please also look on my specific comment related to "Model 7").*

Thank you for the suggestion. We have removed the statement. However, we did find that the refreezing component did not pose significant changes in terms of snow-cover in this approach.

10. *- L 197-198: "falling on the snowpack". While I fully agree that topography (e.g., slope orientation) is important for snowmelt distribution, I would not say it also impacts snowfall temperatures (the shortwave radiation do not much differ between north and south facing*

*slopes during snowfall events). Therefore, I think the Model 4 doesn't make much sense. Nevertheless, I accept authors decision to include it.*

Thank you for the feedback.

11. *- L 257: "grid" instead of "gird".*

    This has been modified.

12. *L 262: I would prefer "seamless" numbering, it means that title "3.1" should follows immediately after title "3". Therefore, I suggest using some title (3.1) for general methodological approach (including Fig. 2 and the list of parameters), continue with title 3.2 named something like "Snow routines variants" (or similar) followed by "3.3 Data requirement of the models" etc.*

    The numbering has been modified in this section.

13. *- Chapter 3.4 would perhaps better fit to discussion.*

    We agree with your statement, but we decided to keep it as it is as we wanted to present before results, how separate calibration helps in reduction of uncertainty. Also regarding the discussion section, we did not want to make it longer, but we have summarized this part in the discussion part.

14. *- L 350 and 362: There are no Figs. 4a and 4b. Or, maybe better put a) to g) labels to individual panels of Fig. 4.*

    This has been revised in the captions.

15. *- Fig. 5: Please add colour scale captions.*

    This figure is omitted from the paper.

16. *- 359: Typo in "efficiency".*

    We wanted to use the term 'efficacy' here.

17. *- L 408: Delete "below" after "Fig. 7" (the figures are placed during post-production and may be placed elsewhere, not necessarily "below").*

    'Below' was omitted.

18. *- Fig. 6: Is the colour scale needed? If I correctly understood, colour scale used here just follows the parameter values, but the parameters are of different physical meaning and different magnitudes thus not comparable to each other. Therefore, I think the colour scale is rather confusing in this context. It would be also good to add units for each parameter. Additionally, please make more informative figure caption. Figures and their captions should be understandable even without the related text. For example, which model variant is shown here? Why the last line represents specific date rather than year as other lines?*

    The figure 6 has been omitted in the revised manuscript. We have tried to make the captions more informative in the revised text.

19. *- Fig. 7: Same as above, please make the Figure caption more informative. Among others, what scores are included within individual plots? Those resulted from 1000 parameter sets? What is represented by the width of individual plots? Please provide clear description in the figure caption. Fig. 8, 9, Table 4 and 5: Same as above, please provide more informative figure caption. For tables, it is not clear what numbers are shown (the fact that it is Brier scores are mentioned only in the text).*

    Figure 7 (now fig. 9) shows the dispersion of Brier scores from the snow-melt model and the HBV snow routine in the different catchments. Figures 8 (now 10) and 9 (now 11) show the parameter ranges and dispersion of NSEs in different catchments. Tables 4 and 5 have been omitted. This information has been added to the respective captions.

20. *- L 425: Typo in "hydrological". Besides, perhaps "Hydrological models validation" would sound better.*

    The typo has been edited.

21. *- L 426-431: This part would fit better to methods section.*

    This is also added in the Methodology section.

22. *- Fig. 10: Why this figure is actually shown here? And why specifically the Horb catchment and the season 2012/13? Please explain it better in the text. I understand that this might be an example to support your conclusion of using separate calibration for snow routines and then for the rest of a hydrological model. However, without any other information it looks like you selected the "best" result to support your conclusion, but without any evidence that also other catchments/years performed similarly well or badly. I would strongly suggest either to put this figure in wider context or remove it. Fig. 10: Y-axis description should contain units.*

    This figure was shown as an illustration of how the approach works in simulating winter flows. However, since other results support the conclusion, we have removed the figure to avoid long manuscript.

23. *- All figures: Besides specific comments above, please check the font size in all figures.*

    We have modified the figure font size.